# Rare Gems: Finding Lottery Tickets at Initialization

**Kartik Sreenivasan**[*w], **Jy-yong Sohn**[*w], **Liu Yang**[w], **Matthew Grinde**[w]
**Alliot Nagle**,[w] **Hongyi Wang**,[c] **Eric Xing**,[mcp] **Kangwook Lee**,[w] **Dimitris Papailiopoulos**[w]

[c] Carnegie Mellon University   [m] Mohamed Bin Zayed University of Artificial Intelligence
[p]Petuum, Inc.   [w] University of Wisconsin-Madison

## Abstract

Large neural networks can be pruned to a small fraction of their original size, with little loss in accuracy, by following a time-consuming "train, prune, re-train" approach. Frankle & Carbin [9] conjecture that we can avoid this by training *lottery tickets*, *i.e.*, special sparse subnetworks found *at initialization*, that can be trained to high accuracy. However, a subsequent line of work [11, 41] presents concrete evidence that current algorithms for finding trainable networks at initialization, fail simple baseline comparisons, *e.g.*, against training random sparse subnetworks. Finding lottery tickets that train to better accuracy compared to simple baselines remains an open problem. In this work, we resolve this open problem by proposing GEM-MINER which finds lottery tickets *at initialization* that beat current baselines. GEM-MINER finds lottery tickets trainable to accuracy competitive or better than Iterative Magnitude Pruning (IMP), and does so up to $19\times$ faster.

## 1   Introduction

A large body of research since the 1980s empirically observed that large neural networks can be compressed or sparsified to a small fraction of their original size while maintaining their predictive accuracy [14–16, 20, 23, 29, 45]. Although several pruning methods have been proposed during the past few decades, many of them follow the "*train, prune, re-train*" paradigm. Although the above methods result in very sparse, accurate models, they typically require several rounds of re-training, which is computationally intensive.

Frankle & Carbin [9] suggest that this computational burden may be avoidable. They conjecture that given a randomly initialized network, one can find *a sparse subnetwork that can be trained to accuracy comparable to that of its fully trained dense counterpart*. This trainable subnetwork found *at initialization* is referred to as a *lottery ticket*. The study above introduced iterative magnitude pruning (IMP) as a means of finding these lottery tickets. Their experimental findings laid the groundwork for what is now known as the *Lottery Ticket Hypothesis* (LTH).

Although Frankle & Carbin [9] establish that the LTH is true for tasks like image classification on MNIST, they were not able to get satisfactory results for more complex datasets like CIFAR-10 and ImageNet when using deeper networks, such as VGG and ResNets [10]. In fact, subsequent work brought the effectiveness of IMP into question. Su et al. [41] showed that even randomly sampled sparse subnetworks at initialization can beat lottery tickets found by IMP as long as the layerwise sparsities are chosen carefully. Gale et al. [12] showed that methods like IMP which train tickets from initialization cannot compete with the accuracy of a model trained with pruning as part of the optimization process.

Frankle et al. [10] explain the failures of IMP using the concept of *linear mode connectivity* which measures the stability of these subnetworks to SGD noise. Extensive follow-up studies propose

---

[*]Authors contributed equally to this paper.

36th Conference on Neural Information Processing Systems (NeurIPS 2022).

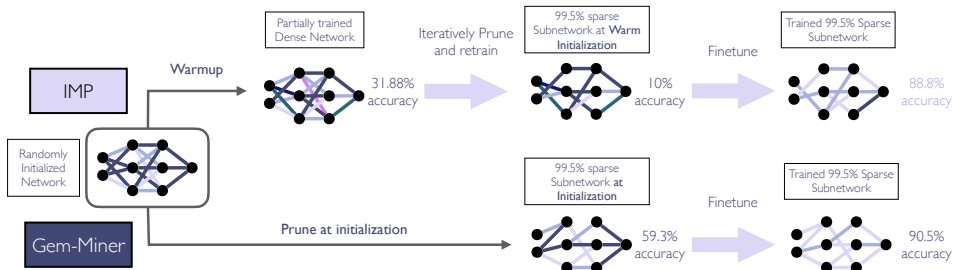

Figure 1: Conceptual visualization of GEM-MINER vs IMP with warmup. The accuracies listed are on a 99.5% sparse VGG-16 trained on CIFAR-10. Given a randomly initialized network, both methods output a subnetwork which is then finetuned. IMP requires warmup *i.e.*, few epochs of training before it can find a sparse subnetwork. GEM-MINER finds a *rare gem*, a subnetwork *at initialization* that achieves high accuracy both before and after weight training.

several heuristics for finding trainable sparse subnetworks at initialization [24, 42, 43]. However, subsequent work by Frankle et al. [11], Su et al. [41] show experimentally that all of these methods fail simple sanity checks. Most methods seem to merely identify good sparsities at each layer, but given those, random sparse subnetworks can be trained to similar or better accuracy.

Frankle et al. [10] show that with a small modification, IMP can beat these sanity checks; the caveat is that it no longer finds these subnetworks at initialization, but after a few epochs of *warm-up* training. Since these subnetworks are found *after initialization*, **IMP with warmup does not find lottery tickets**.

As noted in the original work by Frankle & Carbin [9], the importance of finding trainable subnetworks at initialization is computational efficiency. It is far preferable to train a sparse model from scratch, rather than having to deal with a large dense model, even if that is for a few epochs (which is what IMP with warmup does). To the best of our knowledge, the empirical validity of the *Lottery Ticket Hypothesis*, *i.e.*, the hunt for subnetworks at initialization trainable to SOTA accuracy, remains an open problem.

**Our Contributions.** We resolve this open problem by developing GEM-MINER, an algorithm that finds sparse subnetworks *at initialization*, trainable to accuracy comparable or better than IMP *with warm-up*. GEM-MINER does so by first discovering *rare gems*. Rare gems are subnetworks at initialization that attain accuracy far above random guessing, even before training. Rare gems can then be *refined* to achieve near state-of-the-art accuracy. Simply put, rare gems are lottery tickets that also have high accuracy at initialization.

High accuracy at initialization is not a requirement for a network to be defined as a lottery ticket. However, if our end goal is high accuracy after training, then having high accuracy at initialization likely helps.

Rare gems found by GEM-MINER are the first lottery tickets to beat all baselines in [11, 41]. In Fig. 1 we give a sketch of how GEM-MINER compares with IMP with warm start. GEM-MINER finds subnetworks *at initialization* and is up to 19× faster than IMP which needs warmup.

## 2 Related Work

**Lottery ticket hypothesis.** Following the pioneering work of Frankle & Carbin [9], the search for lottery tickets has grown across several applications, such as language tasks, graph neural networks and federated learning [3, 4, 13, 25]. Savarese et al. [37] propose an alternative to IMP which is significantly faster given enough parallel computing resources. While the LTH itself has yet to be proven mathematically, the so-called strong LTH has been derived which shows that any target network can be approximated by pruning a randomly initialized network with minimal overparameterization [28, 30, 32]. Recently, it has been shown that for such approximation results it suffices to prune a random binary network with slightly larger overparameterization [6, 40].

**Pruning at initialization.** While network pruning has been studied since the 1980s, finding sparse subnetworks at initialization is a more recently explored approach. Lee et al. [24] propose SNIP,

Table 1: We compare the different popular pruning methods in the literature on whether they prune at initialization, are finetunable and pass sanity checks. We also list the amount of computation they need to find a 98.6% sparse subnetwork on ResNet-20, CIFAR-10. For consistency, we do not include the time required to finetune this subnetwork to full accuracy as it would be equal for all methods. For single-shot pruning method we list it as 1 epoch but this depends on the choice of batch-size. Learning Rate Rewinding which we label Renda et al. [34] is a pruning after training algorithm and just outputs a high accuracy subnetwork and hence the sanity checks do not apply to it.

| Pruning Method | Prunes at initialization | Finetunable | Passes sanity checks | Computation to reach 98.6% sparsity |
|---|---|---|---|---|
| IMP [9] | ✗ | ✓ | ✓ | 2850 epochs |
| SNIP [24] | ✓ | ✓ | ✗ | 1 epoch |
| GraSP [43] | ✓ | ✓ | ✗ | 1 epoch |
| SynFlow [42] | ✓ | ✓ | ✗ | 1 epoch |
| Edge-popup [33] | ✓ | ✗ | ✗ | 150 epochs |
| Smart Ratio [41] | ✓ | ✓ | – | $\mathcal{O}(1)$ |
| Learning Rate Rewinding [34] | ✗ | – | – | 3000 epochs |
| **Gem-Miner** | ✓ | ✓ | ✓ | **150 epochs** |

which prunes based on a heuristic that approximates the importance of a connection. Tanaka et al. [42] propose SynFlow which prunes the network to a target sparsity without ever looking at the data. Wang et al. [43] propose GraSP which computes the importance of a weight based on the Hessian gradient product. Patil & Dovrolis [31] propose PHEW which is based on the decomposition of the Neural Tangent Kernel. Lubana & Dick [26] create an interesting theoretical framework based on gradient flow that justifies the successes and failures of several of these algorithms. The goal of these algorithms is to find a subnetwork that can be trained to high accuracy. Ramanujan et al. [33] propose Edge-Popup (EP) which finds a subnetwork at initialization that has high accuracy to begin with. Unfortunately, they also note that in most cases, these subnetworks are not conducive to further finetuning.

The above algorithms are all based on the idea that one can assign a "score" to each weight to measure its importance. Once such a score is assigned, one simply keeps the top fraction of these scores based on the desired target sparsity. This may be done by sorting the scores layer-wise or globally across the network. Additionally, this can be done in *one-shot* (SNIP, GraSP) or *iteratively* (SynFlow). Note that IMP can also be fit into the above framework by defining the "score" to be the *magnitude* of the weights and then pruning globally across the network iteratively.

More recently, Alizadeh et al. [1] propose ProsPr which utilizes the idea of *meta-gradients* through the first few steps of optimization to determine which weights to prune. Their intuition is that this will lead to masks at initialization that are more amenable to training to high accuracy. While it finds high accuracy subnetworks, we show in Section 4.2 that it fails to pass the sanity checks in [11, 41].

**Sanity checks for lottery tickets.** A natural question that arises with pruning at initialization is whether these algorithms are truly finding interesting and nontrivial subnetworks, or if their performance after finetuning can be matched by simply training equally sparse, yet random subnetworks. Ma et al. [27] propose more rigorous definitions of winning tickets and study IMP under several settings with careful tuning of hyperparameters. Frankle et al. [11] and Su et al. [41] introduce several sanity checks (i) Random shuffling (ii) Weight reinitialization (iii) Score inversion and (iv) Random Tickets. Even at their best performance, they show that SNIP, GraSP and SynFlow merely find a good sparsity ratio in each layer and fail to surpass, in term of accuracy, fully trained randomly selected subnetworks, whose sparsity per layer is similarly tuned. Frankle et al. [11] show through extensive experiments that none of these methods show accuracy deterioration after random reshuffling. We explain the sanity checks in detail in Section 4 and use them as baselines to test our own algorithm.

**Pruning during/after training.** While the above algorithms prune at/near initialization, there exists a rich literature on algorithms which prune during/after training. Unlike IMP, algorithms in this category do not rewind the weights. They continue training and pruning iteratively. Frankle et al. [11] and Gale et al. [12] show that pruning at initialization cannot hope to compete with these algorithms. While they do not find lottery tickets, they do find high accuracy sparse networks. Zhu & Gupta [45] propose a gradual pruning schedule where the smallest fraction of weights are pruned at a predefined frequency. They show that this results in models up to 95% sparsity with negligible loss in performance on language as well as image processing tasks. Gale et al. [12] and Frankle et al. [11] also study this as a baseline under the name *magnitude pruning after training*. Renda et al. [34] show that rewinding the learning rate as opposed to weights(like in IMP) leads to the best performing

sparse networks. The closest among these to GEM-MINER is Movement Pruning by Sanh et al. [36] which also computes the mask as a quantized version of the scores in its soft variant. However, it is important to remark that these algorithms do not find Lottery Tickets, merely high accuracy sparse networks. We contrast these different methods in Table 1 in terms of whether they prune at initialization, their finetunability, whether they pass sanity checks as well as their computational costs.

Finally, we note that identifying a good pruning mask can be thought of as training a binary network where the loss is computed over the element-wise product of the original network with the mask. This has been explored in the quantization during training literature [5, 19, 38].

# 3 GEM-MINER: Discovering Rare Gems

**Setting and notation.** Let $S = \{(\boldsymbol{x}_i, y_i)\}_{i=1}^n$ be a given training dataset for a $k$-classification problem, where $\boldsymbol{x}_i \in \mathbb{R}^{d_0}$ denotes a feature vector and label $y_i \in \{1, \ldots, k\}$ denotes its label. Typically, we wish to train a neural network classifier $f(\boldsymbol{w}; \boldsymbol{x}) : \mathbb{R}^{d_0} \to \{1, \ldots, k\}$, where $\boldsymbol{w} \in \mathbb{R}^d$ denotes the set of weight parameters of this neural network. The goal of a pruning algorithm is to extract a mask $\boldsymbol{m} = \{0, 1\}^d$, so that the pruned network is denoted by $f(\boldsymbol{w} \odot \boldsymbol{m}; \boldsymbol{x})$, where $\odot$ denotes the element-wise product. We define the *sparsity* of this network to be the fraction of weights that have been pruned: $s = (1 - \|\boldsymbol{m}\|_0/d)$ following the convention set by Frankle et al. [11]. The loss of a classifier on a single sample $(\boldsymbol{x}, y)$ is denoted by $\ell(f(\boldsymbol{w} \odot \boldsymbol{m}; \boldsymbol{x}), y)$, which captures a measure of discrepancy between prediction and reality. In what follows, we will denote by $\boldsymbol{w}_0 \in \mathbb{R}^d$ to be the set of random initial weights. The type of randomness will be explicitly mentioned when necessary.

**On the path to rare gems; first stop: Maximize pre-training accuracy.** A rare gem needs to satisfy three conditions: (i) sparsity, (ii) non-trivial pre-training accuracy, and (iii) that it can be finetuned to achieve accuracy close to that of the fully trained dense network. This is not an easy task as we have two different objectives in terms of accuracy (pre-training and post-training), and it is unclear if a good subnetwork for one objective is also good for the other. However, since pre-training accuracy serves as a lower bound on the final performace, we focus on maximizing that first, and then attempt to further improve it by finetuning.

Our algorithm is inspired by Edge-Popup (EP) [33]. EP successfully finds subnetworks with high pre-training accuracy but it has two major limitations: (i) it does not work well in the high sparsity regime (*e.g.*, $> 95\%$), and (ii) most importantly, the subnetworks it finds are typically not conducive to further finetuning in that the final accuracy does not approach the performance of IMP.

In the following, we take GEM-MINER apart and describe the components that allow it to surpass these issues.

**GEM-MINER without sparsity control.** Much like EP, GEM-MINER employs a form of backpropagation, and works as follows. Each of the random weights $[\mathbf{w}_0]_i$ in the original network is associated with a normalized score $p_i \in [0, 1]$. These normalized scores become our optimization variables and are responsible

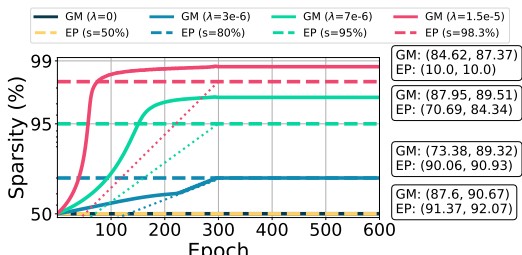

Figure 2: The sparsity of intermediate results, the accuracy of the final output, and the accuracy after finetuning on MobileNet-V2, CIFAR-10. For GEM-MINER (GM), we also visualize the sparsity upper bounds as dotted lines. As $\lambda$ increases, note that the sparsity of GEM-MINER's output increases. For $\lambda = 3 \cdot 10^{-6}$, the iterative freezing algorithm kicks in around epoch 220, regularizing the sparsity thereafter. The gem found by GEM-MINER($\lambda = 1.5 \cdot 10^{-5}$) achieves an accuracy of 84.62% before finetuning and 87.37% after finetuning, while EP is unable to achieve non-trivial accuracy before or after finetuning at 98.3% sparsity.

for computing the *supermask* $\mathbf{m}$, *i.e.*, the pruning pattern of the network at initialization.

For a given set of weights $\boldsymbol{w}$ and scores $\boldsymbol{p}$, GEM-MINER sets the effective weights as $\boldsymbol{w}_{\text{eff}} = \boldsymbol{w} \odot r(\boldsymbol{p})$, where $\text{r}(\cdot)$ is an element-wise rounding function, and $\mathbf{m} = r(\boldsymbol{p})$ is the resulting supermask. The rounding function can be changed, *e.g.*, $r$ can perform randomized rounding, in which case $p_i$ would be the probability of keeping weight $w_i$ in $\mathbf{m}$. In our case, we found that simple deterministic rounding, *i.e.*, $r(p_i) = \mathbf{1}_{p_i \geq 0.5}$ works well.

At every iteration GEM-MINER samples a batch of training data and performs backpropagation on the loss of the effective weights, with respect to the scores $p$, while projecting back to $[0, 1]$ when needed. During the forward pass, due to the rounding function, the effective network used is indeed a subnetwork of the given network. Here, since $r(p)$ is a non-differentiable operation we use the Straight Through Estimator (STE) [2] which backpropagates through the indicator function as though it were the identity function. Therefore, even pruned scores can receive non-zero gradients which allows them to revive over the course of training.

Note that this vanilla version of GEM-MINER is unable to exercise control over the final sparsity of the model. For reasons that will become evident in below, we will call this version of our algorithm GEM-MINER(0). There is already a stark difference from EP: GEM-MINER(0) will automatically find the optimal sparsity, while EP requires the target sparsity $s$ as an input parameter.

However, at the same time, this also significantly limits the applicability of GEM-MINER(0) as one cannot obtain a highly sparse gem. Shown as a dark blue curve in Fig. 2 is the sparsity of GEM-MINER(0). Here, we run GEM-MINER with a randomly initialized MobileNet-V2 network on CIFAR-10. Note that the sparsity stays around 50% throughout the run, which is consistent with the observation by Ramanujan et al. [33] that accuracy of subnetworks at initialization is maximized at around 50% sparsity.

---

**Algorithm 1: GEM-MINER**

**Input:** Dataset $D = \{(x_i, y_i)\}$, learning rate $\eta$, rounding function $r(\cdot)$, number of epochs $E$, freezing period $T$, target sparsity $s \in [0, 1]$

**Output:** Mask $m = r(p) \odot q \in \{0, 1\}^d$

1   $c \leftarrow \frac{1}{E} \ln\left(\frac{1}{1-s}\right), q \leftarrow \mathbf{1}_d$

2   $w, p \leftarrow$ random vector in $\mathbb{R}^d$,

3           random vector in $[0, 1]^d$

4   **for** $j$ in $1, 2, \ldots, E$ **do**

5       **for** $(x_i, y_i) \in D$ **do**

6           $w_{\text{eff}} \leftarrow (w \odot q) \odot r(p)$

7           $p \leftarrow p - \eta \nabla_p \ell(f(w_{\text{eff}}; x_i), y_i)$

8           /* STE */

9           $p \leftarrow \text{proj}_{[0,1]^d} p$

10       **if** $\text{mod}(j, T) = 0$ **then**

11           $I_1 \leftarrow \{i : q_i = 1\}$

12           $p_{sorted} \leftarrow \text{sort}(p_{i \in I_1})$

13           $p_{bottom} \leftarrow$ Bottom-$(1 - e^{-cj})$ fraction

14               of $p_{sorted}$

15           $q \leftarrow q \odot \mathbb{1}_{p_i \notin p_{bottom}}$

---

**Regularization and Iterative freezing.** GEM-MINER(0) is a good baseline algorithm for finding accurate subnetworks at initialization, but it cannot be used to find *rare gems*, which need to be sparse and trainable. To overcome this limitation, we apply a standard trick – we add a regularization term to encourage sparsity. Thus, in addition to the task loss computed with the effective weights, we also compute the $L_2$ or $L_1$ norm of the score vector $p$ and optimize over the total regularized loss. More formally, we minimize $\ell := \ell_{\text{task}} + \lambda \ell_{\text{reg}}$, where $\lambda$ is the hyperparameter and $\ell_{\text{reg}}$ is either $L_2$ or $L_1$ norm of the score vector $p$.

We call this variant GEM-MINER($\lambda$), where $\lambda$ denotes the regularization weight. This naming convention should explain why we called the initial version GEM-MINER(0).

The experimental results in Fig. 2 show that this simple modification indeed allows us to control the sparsity of the solution. We chose to use the $L_2$ regularizer, however preliminary experiments showed that $L_1$ performs almost identically. By varying $\lambda$ from $\lambda = 0$ to $\lambda = 7 \cdot 10^{-6}$ and $\lambda = 1.5 \cdot 10^{-5}$, the final sparsity of the gem found by GEM-MINER($\lambda$) becomes $97.5\%$ and $98.6\%$, respectively.

One drawback of this regularization approach is that it only indirectly controls the sparsity. If we have a target sparsity $s$, then there is no easy way of finding the appropriate value of $\lambda$ such that the resulting subnetwork is $s$-sparse. If we choose $\lambda$ to be too large, then it will give us a gem that is way too sparse; too small a $\lambda$ and we will end up with a denser gem than what is needed. As a simple heuristic, we employ *iterative freezing*, which is widely used in several existing pruning algorithms, including IMP [9, 12, 45]. More specifically, we can design an exponential function $\overline{s}(j) = 1 - e^{-cj}$ for some $c > 0$, which will serve as the upper bound on the sparsity. If the total number of epochs is $E$ and the target sparsity is $s$, we have $\overline{s}(E) = 1 - e^{-cE} = s$. Thus, we have $c = \frac{1}{E} \ln\left(\frac{1}{1-s}\right)$.

Once this sparsity upper bound is designed, the iterative freezing mechanism regularly checks the current sparsity to see if the lower bound is violated or not. If the sparsity bound is violated, it finds the smallest scores, zeros them out, and freezes their values thereafter. By doing so, we can guarantee the final sparsity even when $\lambda$ was not sufficiently large. To see this freezing mechanism in action, refer the blue curve in Fig. 2. Here, the sparsity lower bounds (decreasing exponential functions) are visualized as dotted lines. Note that for the case of $\lambda = 3 \cdot 10^{-6}$, the sparsity of the network does

not decay as fast as desired, so it touches the sparsity lower bound around epoch 220. The iterative freezing scheme kicks in here, and the sparsity decay is controlled by the lower bound thereafter, achieving the specified target sparsity at the end.

The full pseudocode of GEM-MINER is provided in Algorithm 1. There are two minor implementation details which differ from the explanation above: (i) we impose the iterative freezing every $T$ epochs, not every epoch and (ii) iterative freezing is imposed even when the sparsity bound is not violated.

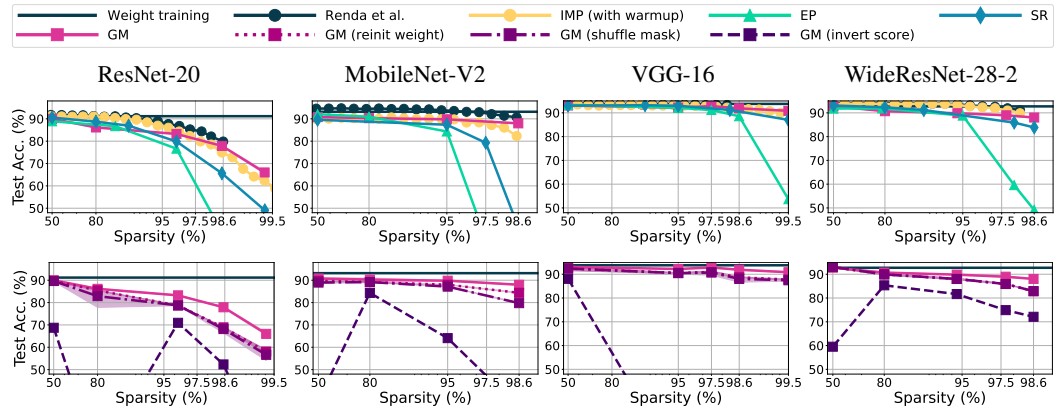

Figure 3: Performance of different pruning algorithms on CIFAR-10 for benchmark networks. Top: post-finetune accuracy; Bottom: sanity check methods suggested in Frankle et al. [11] applied on GEM-MINER (GM). Note that GM achieves similar post-finetune accuracy as IMP, and typically outperforms it in the sparse regime. GM has higher post-finetune accuracy than EP and Smart Ratio (SR). GM also passes the sanity checks suggested in Frankle et al. [11]. Finally, GM (which prunes *at* init) nearly achieves the performance of Renda et al. (which is a pruning after training method) in the sparse regime, e.g., 98.6% sparsity in ResNet-20.

## 4  Experiments

In this section, we present the experimental results[2] for the performance of GEM-MINER across various tasks.

**Tasks.**  We evaluate our algorithm on **(Task 1)** CIFAR-10 classification, on ResNet-20, MobileNet-V2, VGG-16, and WideResNet-28-2, **(Task 2)** TinyImageNet classification on ResNet-18 and ResNet-50, **(Task 3)** Finetuning on the Caltech-101 [7] dataset using a ResNet-50 pretrained on ImageNet, and **(Task 4)** CIFAR-100 classification using ResNet-32. The detailed description of the datasets, networks and hyperparameters can be found in Section A of the Appendix.

**Proposed scheme.**  We run GEM-MINER with an $L_2$ regularizer. If a network reaches its best accuracy after $E$ epochs of dense training, then we run GEM-MINER for $E$ epochs from random init to get a sparse subnetwork *at initialization*, and then run weight training on the sparse subnetwork for another $E$ epochs.

**Comparisons.**  We tested our method against the following baselines: dense weight training and four pruning algorithms: (i) IMP [10], (ii) Learning rate rewinding [34], denoted by Renda et al., (iii) Edge-Popup (EP) [33], and (iv) Smart-Ratio (SR) which is the random pruning method proposed by Su et al. [41].

We also ran the following sanity checks, proposed by Frankle et al. [11]: (i) (Random shuffling): To test if the algorithm prunes specific connections, we randomly shuffle the mask at every layer. (ii) (Weight reinitialization): To test if the final mask is specific to the weight initialization, we reinitialize the weights from the original distribution. (iii) (Score inversion): Since most pruning algorithms use a heuristic/score function as a proxy to measure the importance of different weights, we invert the scoring function to check whether it is a valid proxy. More precisely, this test involves pruning the weights which have the *smallest* scores rather than the largest. In all of the above tests, if the accuracy after finetuning the new subnetwork does not deteriorate significantly, then the algorithm is merely identifying optimal layerwise sparsities.

---

[2]Our codebase can be found at `https://github.com/ksreenivasan/pruning_is_enough`.

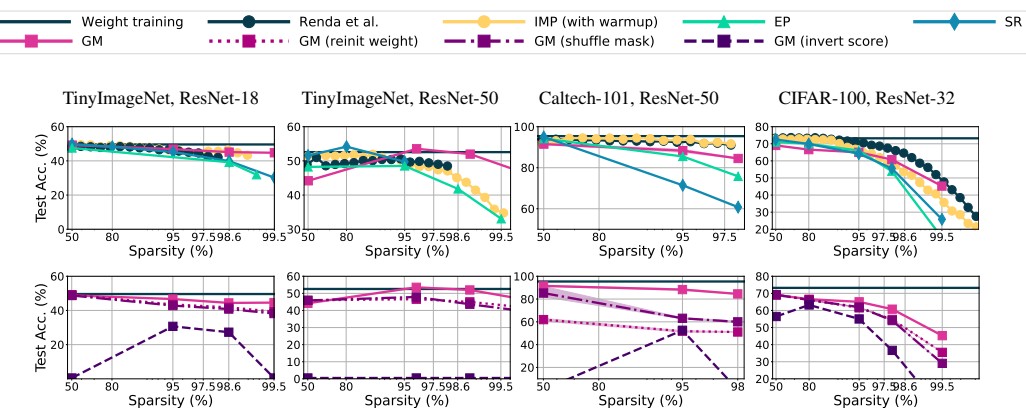

Figure 4: Accuracy on image classification tasks on TinyImageNet, Caltech-101 and CIFAR-100. For Caltech-101, we pruned a pre-trained ImageNet model (ResNet-50). Top: post-finetune accuracy, bottom: sanity check methods suggested in Frankle et al. [11] applied on GEM-MINER.

## 4.1 Rare gems obtained by GEM-MINER

**Task 1.** Fig. 3 shows the sparsity-accuracy tradeoff for various pruning methods trained on CIFAR-10 using ResNet-20, MobileNet-V2, VGG-16 and WideResNet-28-2. For each column (network), we compare IMP, IMP with learning rate rewinding (Renda et al.), GEM-MINER, EP, and SR in two performance metrics: the top row shows the accuracy of the subnetwork after weight training and bottom row shows the result of the sanity checks on GEM-MINER.

As shown in the top row of Fig. 3, GEM-MINER finds a lottery ticket *at* initialization. It reaches accuracy similar to IMP after weight training. Moreover, in the sparse regime (e.g., above 98.6% for ResNet-20 and MobileNet-V2), GEM-MINER outperforms IMP in terms of post-finetune accuracy. The bottom row of Fig. 3 shows that GEM-MINER passes the sanity check methods. For all networks, the performance in the sparse regime (98.6% sparsity or above) shows that the suggested GEM-MINER algorithm enjoys 3–10% accuracy gap with the best performance among variants. The results in the top row show that GEM-MINER far outperforms the random network with smart ratio (SR).

**Tasks 2–4.** Fig. 4 shows the sparsity-accuracy tradeoff for Tasks 2–4. Similar to Fig. 3, the top row reports the accuracy *after* weight training, and the bottom row contains the results of the sanity checks.

As shown in Fig. 4a and Fig. 4b, the results for **Task 2** show that (i) GEM-MINER achieves accuracy comparable to IMP as well as Renda et al. (IMP with learning rate rewinding) even in the sparse regime, (ii) GEM-MINER has non-trivial accuracy before finetuning (iii) GEM-MINER passes all the sanity checks, and (iv) GEM-MINER outperforms EP and SR. These results show that GEM-MINER successfully finds rare gems even in the sparse regime for **Task 2**.

Fig. 4c shows the result for **Task 3**. Unlike other tasks, GEM-MINER does not reach the post-finetune accuracy of IMP, but GEM-MINER enjoys over an 8% accuracy gap compared with EP and SR. Moreover, the bottom row shows that GEM-MINER has over 20% higher accuracy than the sanity checks above 95% sparsity showing that the subnetwork found by GEM-MINER is unique in this sparse regime.

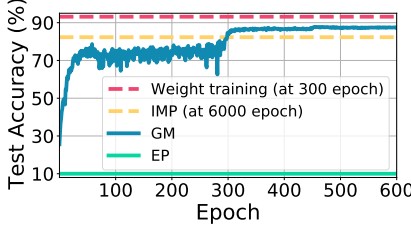

Figure 5: Convergence plot for CIFAR-10, MobileNet-V2 experiments, where we apply GEM-MINER for 300 epochs and then finetune the sparse model for another 300 epochs, to reach 98.6% sparse model. We include the accuracy of dense weight training, IMP and EP (98.6% sparse model) as references. Note that the comparison is with 300 epochs of weight training, and IMP using 20 rounds of iterative pruning, i.e., 300 × 20 = 6000 epochs, to reach 98.6% sparsity. GEM-MINER achieves a higher accuracy than IMP despite its 19× shorter runtime to find a sparse subnetwork.

Fig. 4d shows the result for **Task 4** where once again, GEM-MINER is comparable to IMP throughout and outperforms it in the sparse regime.

Table 2: We compare ProsPr [1] vs GEM-MINER on ResNet-20, CIFAR-10 and run the random shuffling as well as the weight reinit sanity checks. Note that GEM-MINER produces a subnetwork that is higher accuracy despite being more sparse. Moreover, ProsPr does not show significant decay in performance after the sanity checks while GEM-MINER does. Therefore, it is likely that ProsPr is merely identifying good layerwise sparsity ratios.

| Algorithm | Sparsity | Accuracy after finetune | Accuracy after Random shuffling | Accuracy after Weight reinitialization |
|---|---|---|---|---|
| ProsPr | 95% | 82.67% | 82.15% | 81.64% |
| **GEM-MINER** | **96.28%** | **83.4%** | **78.73%** | **78.6%** |

## 4.2 Comparison to ProsPr

Alizadeh et al. [1] recently proposed a pruning at init method called ProsPr which utilizes meta-gradients through the first few steps of optimization to determine which weights to prune, thereby accounting for the "trainability" of the resulting subnetwork. In Table 2 we compare it against GEM-MINER on ResNet-20, CIFAR-10 and also run the (i) Random shuffling and (ii) Weight reinitialization sanity checks from Frankle et al. [11]. We were unable to get ProsPr using their publicly available codebase to generate subnetworks at sparsity above 95% and therefore chose that sparsity. Note that GEM-MINER produces a subnetwork that is higher accuracy despite being more sparse. After finetuning for 150 epochs, our subnetwork reaches 83.4% accuracy while the subnetwork found by ProsPr only reaches 82.67% after training for 200 epochs. More importantly, ProsPr does not show significant decay in performance after the random reshuffling or weight reinitialization sanity checks. Therefore, as Frankle et al. [11] remark, it is likely that it is identifying good layerwise sparsity ratios, rather than a mask specific to the initialized weights.

## 4.3 Observations on GEM-MINER

**Convergence of accuracy and sparsity.** Fig. 5 shows how the accuracy of GEM-MINER improves as training progresses, for MobileNet-V2 on CIFAR-10 at sparsity 98.6%. This shows that GEM-MINER, reaches high accuracy even early in training, and can be finetuned to accuracy higher than that of IMP (which requires $19\times$ the runtime than our algorithm). EP remains at random-guessing throughout as it typically does not work well in the sparse regime.

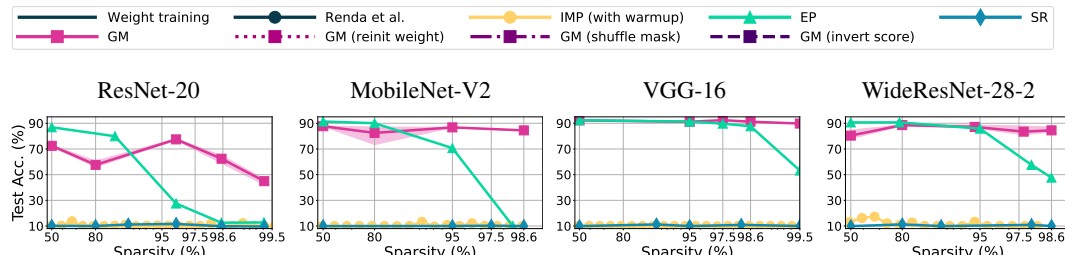

Figure 6: Performance of different pruning algorithms before finetuning on CIFAR-10 for benchmark networks. GEM-MINER finds subnetworks that already have reasonably high accuracy even before weight training. Note that, while IMP and SR have scarcely better than random guessing at initialization, subnetworks found by GEM-MINER typically perform even better than EP, especially in the sparse regime.

**High pre-finetune accuracy.** As shown in Fig. 6, GEM-MINER finds subnetworks at initialization that have a reasonably high accuracy even before the weight training, e.g., above 90% accuracy for 98.6% sparsity in VGG-16, and 85% accuracy for 98.6% sparsity in MobileNet-V2. Note that, in contrast, IMP and SR have accuracy scarcely better than random guessing at initialization. Clearly, GEM-MINER fulfills its objective in maximizing accuracy before finetuning and therefore finds rare gems – lottery tickets at initialization which already have high accuracy.

**Limitations of GEM-MINER.** We observed that in the dense regime (50% sparsity, 80% sparsity), GEM-MINER sometimes performs worse than IMP. While we believe that this can be resolved by appropriately tuning the hyperparameters, we chose to focus our attention on the sparse regime. We would also like to remark that GEM-MINER is fairly sensitive to the choice of hyperparameters and for some models, we had to choose different hyperparameters for each sparsity to ensure optimal performance. Though this occurs rarely, we also find that an extremely aggressive choice of $\lambda$ can

Table 3: We construct different variants of EP and compare their performance with GEM-MINER, for ResNet-20, CIFAR-10, 99.41% sparsity. We establish that having a global score metric and gradually pruning is key to improved performance.

| Pruning Method | EP | Global EP | Global EP with Gradual Pruning | Global EP with Gradual Pruning and Regularization | GEM-MINER |
|---|---|---|---|---|---|
| **Pre-finetune acc (%)** | 19.57 | 22.22 | 31.56 | 19.67 | **45.30** |
| **Post-finetune acc (%)** | 24.47 | 34.42 | 63.54 | 63.72 | **66.15** |

lead to *layer-collapse* where one or more layers gets pruned completely. This happens when all the scores $p$ of that layer drop below 0.5.

**Layer-wise sparsity.** We compare the layer-wise sparsities of different algorithms for ResNet-20 on CIFAR-10 in Fig. 7. Both GEM-MINER and IMP spend most of their sparsity budget on the first and last layers. SR assigns 70% sparsity to the last layer and the sparsity increases smoothly across the others. EP maintains the target sparsity ratio at each layer and therefore is always a horizontal line.

**How does GEM-MINER resolve EP's failings?** An open problem from Ramanujan et al. [33] is why most subnetworks found by EP are not fine-tunable. While GEM-MINER is significantly different from EP, it is reasonable to ask which modification allowed it to find lottery tickets without forgoing high accuracy at initialization. Table 3 explores this question for ResNet-20, CIFAR-10 at 99.41% sparsity. Here, we compare EP, GEM-MINER, as well as three EP vari-

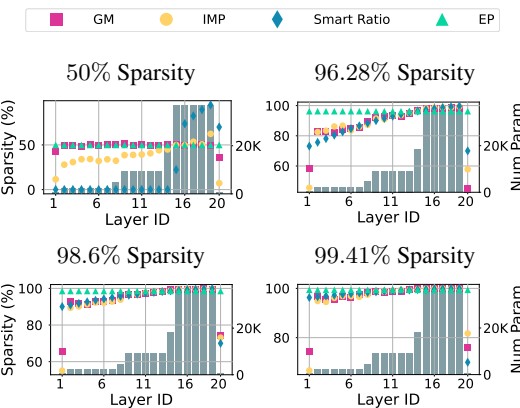

Figure 7: The layerwise sparsity for ResNet-20 pruned by GEM-MINER, IMP, Smart Ratio, and EP. The dark bar is the layerwise number of parameters. Both GEM-MINER and IMP utilize most of the sparsity budget for the first and last layers.

ants that we construct. (i) *(Global EP)* is a modification where the bottom-$k$ scores are pruned globally, not layer-wise. This allows the algorithm to trade-off sparsity in one layer for another. (ii) *(Gradual pruning)* reduces the parameter $k$ gradually as opposed to setting it to the target sparsity from the beginning. (iii) *(Regularization)*: we add an $L_2$ term on the score $p$ of the weights to encourage sparsity. The results indicate that *global pruning* and *gradual pruning* significantly improve both the pre and post-finetune accuracies of EP. Adding regularization does not improve the performance significantly. Finally, adding all three features to EP allows it to achieve 63.72% accuracy, while GEM-MINER reaches 66.15% accuracy. It is important to note that even with all three features, EP is inherently different from GEM-MINER in how it computes the *supermask* based on the scores. But we conjecture that aggressive, layerwise pruning is the key reason for EP's failings.

**Ablation study on GEM-MINER.** To better understand the relative importance of the different components of the algorithm, we do a more thorough ablation study in the same vein as the above analysis. We consider the setting of ResNet-20, CIFAR-10 at a sparsity of 98.56% and evaluate the performance of (i) GEM-MINER, (ii) GEM-MINER without regularization and (iii) GEM-MINER without regularization and without global pruning. As shown in Table 4, GEM-MINER outperforms all of its ablated versions. In fact, GEM-MINER without regularization works extremely poorly, and the final variant is barely above random guessing. Note that when we ablate global pruning, the algorithm chooses the bottom$-k$ weights in each layer much like EP.

Table 4: Ablation study of GEM-MINER on ResNet-20, CIFAR-10 at 98.56% sparsity. GM outperforms all of its variants and in fact, when we ablate regularization and global pruning, the performance is barely above random guessing. (GM - Regularization) denotes GM without regularization.

| GEM-MINER variant | Accuracy before FT (%) | Accuracy after FT (%) |
|---|---|---|
| **GM** | **61.23** | **77.12** |
| GM - Regularization | 10.18 | 27.41 |
| GM - Regularization - Global Pruning | 10.08 | 11.6 |

Table 5: Comparison of GEM-MINER and its longer version, for ResNet-20, CIFAR-10 at 98.6% sparsity. LONG GM, when given the same number of epochs rivals the performance of Renda et al. [34]

| Method | GM (cold) | Long GM (cold) | IMP (warm) | Renda et al. (pruning after training) |
|---|---|---|---|---|
| **Number of Epochs** | 300 | 3000 | 3000 | 3000 |
| **Accuracy (%)** | 77.89 | 79.50 | 74.52 | 80.21 |

**Applying GEM-MINER for longer periods.** Recall that GEM-MINER uses $19 \times$ fewer epochs than methods like IMP [10] and Learning rate rewinding (Renda et al. [34]), to find a subnetwork at 98.6% sparsity which can then be trained to high accuracy. Here, we consider a long version of GEM-MINER to see if it can benefit if it is allowed to run for longer. Table 5 shows the comparison of post-finetune accuracy for GEM-MINER, LONG GEM-MINER, IMP and Renda et al. [34] tested on ResNet-20, CIFAR-10 at 98.5% sparsity. Regular GEM-MINER, applies iterative freezing every 5 epochs to arrive at the target sparsity in 150 epochs. LONG GEM-MINER instead prunes every 150 epochs and therefore reaches the target sparsity in 3000 epochs. We find that applying GEM-MINER for longer periods improves the post-finetune accuracy in this regime by 1.5%. This shows that given equal number of epochs, GEM-MINER, which prunes at initialization, can close the gap to Learning rate rewinding [34] which is a prune-after-training method.

## 5 Conclusion

In this work, we resolve the open problem of pruning at initialization by proposing GEM-MINER, an algorithm that finds *rare gems* – lottery tickets *at initialization* that have non-trivial accuracy even before finetuning, and accuracy rivaling prune-after-train methods after finetuning. Unlike other methods, subnetworks found by GEM-MINER pass all known sanity checks and baselines. Moreover, we show that GEM-MINER is competitive with IMP despite not using warmup and up to $19\times$ faster.

## Acknowledgements

The authors would like to thank Jeff Linderoth for early discussions on viewing pruning as an integer programming problem. This research was supported by ONR Grant No. N00014-21-1-2806 and NSF/Intel Partnership on Machine Learning for Wireless Networking Program under Grant No. CNS-2003129.

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
