# Contents of the Appendix

## A  Experimental Setup

In this section, we introduce the datasets (A.1) and models (A.2) that we used in the experiments. We also report the detailed hyperparameter choices (A.3) of GEM-MINER each network and sparsity level. For competing methods, we used hyperparameters used by the original authors whenever possible. In other cases, we tried SGD (with momentum) and Adam optimizers, initial learning rate (LR) of $\eta$, $0.1\eta$, $10\eta$, and cosine/multi- step LR decay, where $\eta$ is the best LR for weight training. All of our experiments are run using PyTorch 1.11 on Nvidia 3090 TIs and Nvidia V100s.

### A.1  Dataset

In the experiments, we demonstrate the performance of GEM-MINER across various datasets. For each dataset, we optimize the training loss, and tune hyperparameters based on the validation accuracy. The test accuracy is reported for the best model chosen based on the validation accuracy.

**CIFAR-10.**  CIFAR-10 consists of $60,000$ images from $10$ classes, each with size $32 \times 32$, of which $50000$ images are used for training, and $10,000$ images are for testing [22]. For data processing, we follow the standard augmentation: normalize channel-wise, randomly horizontally flip, and random cropping. For hyperparameter tuning we randomly split the train set into $45000$ train images and retain $5000$ images as the validation set. Once the hyperparameters are chosen, we retrain on the full train set and report test accuracy.

**TinyImageNet.**  TinyImageNet contains $100000$ images of $200$ classes ($500$ each class), which are downsized to $64 \times 64$ colored images. Each class has $500$ training images, $50$ validation images and $50$ test images. Augmentation includes normalizing, random rotation and random flip. Train set, validation set, and test set are provided.

**Caltech-101.**  Caltech-101 contains figures of objects from $101$ categories. There are around $40$ to $800$ images per category, and most categories have about $50$ images [7]. The size of each image is roughly $300 \times 200$ pixels. When processing the image, we resize each figure to $224 \times 224$, and normalize it across channels. We split $20\%$ of the data to be test set, and in the remaining training set, we retain $25\%$ as the validation set, giving us train/val/test $= 60\%/20\%/20\%$ split.

**CIFAR-100.**  The CIFAR-100 dataset is just like CIFAR-10 except that it has $100$ classes containing $600$ images each. Therefore, it has $60,000$ images from $100$ classes, each with size $32 \times 32$, of which $50000$ images are used for training, and $10,000$ images are for testing [22]. For data processing, we follow the standard augmentation: normalize channel-wise, randomly horizontally flip, and random cropping. For hyperparameter tuning we randomly split the train set into $45000$ train images and retain $5000$ images as the validation set. Once the hyperparameters are chosen, we retrain on the full train set and report test accuracy.

### A.2  Model

Unless otherwise specified, in all of our experiments experiments, we use Non-Affine BatchNorm, and disable bias for all the convolution and linear layers. We find that most implementations of pruning algorithms instead use them and merely ignore them while pruning and while computing sparsity. While they do not alter sparsity by much (since there are few parameters when compared to weights), we still find this to be inaccurate. Moreover, it is not obvious how to prune biases – should they be treated as weight? Should they be treated as a different set of parameters? In order to make sure we compare all the baselines on the same platform, we decided that eliminating them was the fair choice (As [8] describe, pruning with biases is an interesting problem but needs to be handled slightly carefully). We use uniform initialization for scores, signed constant initialization [33] for

weight parameters for GEM-MINER while dense training initializes the weights using the standard Kaiming normal initializaiton [17].

The networks that we used in our experiments are summarized as follows.

**ResNet-18, ResNet-20, ResNet-32 and ResNet-50 [18].**   We follow the standard ResNet archi- tecture. ResNet-20 is designed for CIFAR-10, ResNet-32 is for CIFAR-100, while ResNet-18 and ResNet-50 are for ImageNet. In Table 6, we use the convention [kernel × kernel, output]×(times repeated) for the convolution layers found in ResNet blocks. We base our implementation on the following GitHub repository[3].

**WideResNet-28-2 [44].**   We base our implementation on the following GitHub repository[4]. Archi- tecture details can be found in Table 6.

Table 6: ResNet architecture used in our experiments. The output layer of the network is changed according to the dataset. For example, ResNet50 is used for both TinyImageNet and ImageNet so we changed the output dimension to 200 and 1000 respectively.

| Layer | ResNet-20 | ResNet-18 | ResNet-50 | WideResNet-28-2 |
|---|---|---|---|---|
| Conv 1 | 3×3, 16 padding 1 stride 1 | 3×3, 64 padding 3 stride 2 | 7×7, 64 padding 3 stride 2 | 3×3, 16 padding 1 stride 1 |
| | | Max Pool, kernel size 3, stride 2, padding 1 | | |
| Layer stack 1 | $\begin{bmatrix} 3{\times}3,\ 16 \\ 3{\times}3,\ 16 \end{bmatrix}{\times}3$ | $\begin{bmatrix} 3{\times}3,\ 64 \\ 3{\times}3,\ 64 \end{bmatrix}{\times}2$ | $\begin{bmatrix} 1{\times}1,\ 64 \\ 3{\times}3,\ 64 \\ 1{\times}1,\ 256 \end{bmatrix}{\times}3$ | $\begin{bmatrix} 3{\times}3,\ 32 \\ 3{\times}3,\ 32 \end{bmatrix}{\times}4$ |
| Layer stack 2 | $\begin{bmatrix} 3{\times}3,\ 32 \\ 3{\times}3,\ 32 \end{bmatrix}{\times}3$ | $\begin{bmatrix} 3{\times}3,\ 128 \\ 3{\times}3,\ 128 \end{bmatrix}{\times}2$ | $\begin{bmatrix} 1{\times}1,\ 128 \\ 3{\times}3,\ 128 \\ 1{\times}1,\ 512 \end{bmatrix}{\times}4$ | $\begin{bmatrix} 3{\times}3,\ 64 \\ 3{\times}3,\ 64 \end{bmatrix}{\times}4$ |
| Layer stack 3 | $\begin{bmatrix} 3{\times}3,\ 64 \\ 3{\times}3,\ 64 \end{bmatrix}{\times}3$ | $\begin{bmatrix} 3{\times}3,\ 256 \\ 3{\times}3,\ 256 \end{bmatrix}{\times}2$ | $\begin{bmatrix} 1{\times}1,\ 256 \\ 3{\times}3,\ 256 \\ 1{\times}1,\ 1024 \end{bmatrix}{\times}6$ | $\begin{bmatrix} 3{\times}3,\ 128 \\ 3{\times}3,\ 128 \end{bmatrix}{\times}4$ |
| Layer stack 4 | - | $\begin{bmatrix} 3{\times}3,\ 512 \\ 3{\times}3,\ 512 \end{bmatrix}{\times}2$ | $\begin{bmatrix} 1{\times}1,\ 512 \\ 3{\times}3,\ 512 \\ 1{\times}1,\ 2048 \end{bmatrix}{\times}3$ | - |
| FC | Avg Pool, kernel size 8 64 × N_CLASSES | Adaptive Avg Pool, output size $(1, 1)$ 512 × N_CLASSES      2048 × N_CLASSES | | Avg Pool, kernel size 8 128 × N_CLASSES |

**VGG-16 [39].**   In the original VGG-16 network, there are 13 convolution layers and 3 FC layers (including the last linear classification layer). We follow the VGG-16 architectures used in [10, 11] to remove the first two FC layers while keeping the last linear classification layer. This finally leads to a 14-layer architecture, but we still call it VGG-16 as it is modified from the original VGG-16 architectural design. Detailed architecture is shown in Table 7. We base our implementation on the GitHub repository[5].

**MobileNet-V2 [35].**   We base our implementation on the GitHub repository.[6]   Details of the architecture is shown in Table 8.

### A.3   Hyper-Parameter Configuration

In this section, we will state the hyperparameter configuration for GEM-MINER and finetuning lottery tickets. For each dataset, model and different target sparsity, we tuned our hyperparameters for GEM-MINER by trying out different values of learning rate and L2 regularization weight $\lambda$. We also

---

[3]https://github.com/akamaster/pytorch_resnet_cifar10/blob/master/resnet.py
[4]https://github.com/xternalz/WideResNet-pytorch/blob/master/wideresnet.py
[5]https://github.com/kuangliu/pytorch-cifar/blob/master/models/vgg.py
[6]https://github.com/kuangliu/pytorch-cifar/blob/master/models/mobilenetv2.py

Table 7: Detailed architecture of the VGG-16 architecture used in our experiments. We have a non-affine batchnnorm layer followed by a ReLU activation after each convolutional layer (omitted in the table). The shapes for convolution layers follow $(c_{in}, c_{out}, k, k)$.

| Parameter | Shape | Layer hyper-parameter |
|---|---|---|
| layer1.conv1.weight | $3 \times 64 \times 3 \times 3$ | stride:1;padding:1 |
| layer2.conv2.weight | $64 \times 64 \times 3 \times 3$ | stride:1;padding:1 |
| pooling.max | N/A | kernel size:2;stride:2 |
| layer3.conv3.weight | $64 \times 128 \times 3 \times 3$ | stride:1;padding:1 |
| layer4.conv4.weight | $128 \times 128 \times 3 \times 3$ | stride:1;padding:1 |
| pooling.max | N/A | kernel size:2;stride:2 |
| layer5.conv5.weight | $128 \times 256 \times 3 \times 3$ | stride:1;padding:1 |
| layer6.conv6.weight | $256 \times 256 \times 3 \times 3$ | stride:1;padding:1 |
| layer7.conv7.weight | $256 \times 256 \times 3 \times 3$ | stride:1;padding:1 |
| pooling.max | N/A | kernel size:2;stride:2 |
| layer8.conv9.weight | $256 \times 512 \times 3 \times 3$ | stride:1;padding:1 |
| layer9.conv10.weight | $512 \times 512 \times 3 \times 3$ | stride:1;padding:1 |
| layer10.conv11.weight | $512 \times 512 \times 3 \times 3$ | stride:1;padding:1 |
| pooling.max | N/A | kernel size:2;stride:2 |
| layer11.conv11.weight | $512 \times 512 \times 3 \times 3$ | stride:1;padding:1 |
| layer12.conv12.weight | $512 \times 512 \times 3 \times 3$ | stride:1;padding:1 |
| layer13.conv13.weight | $512 \times 512 \times 3 \times 3$ | stride:1;padding:1 |
| pooling.max | N/A | kernel size:2;stride:2 |
| pooling.avg | N/A | kernel size:1;stride:1 |
| layer14.conv14.weight | $512 \times 10 \times 1 \times 1$ | stride:1;padding:1 |

test different pruning periods of 5, 8, and 10 epochs. Finally, we also tried ADAM [21] and SGD. While SGD usually comes out on top, there were some settings where ADAM performed better.

### A.3.1 GEM-MINER Training

We tested the CIFAR-10 dataset on the following architectures: i) ResNet-20 ii) MobileNet-V2 iii) VGG-16 iv) WideResNet-28-2. For TinyImageNet, we test on the architectures: i) ResNet-18 ii) ResNet-50. We tested the transfer learning on pretrained ImageNet model, where the target task is classification on Caltech-101 dataset with 101 classes. We first loaded the ResNet-50 model pretrained for ImageNet[7] and changed the last layer to a single fully-connected network having size $2048 \times 101$. To match the performance of the pretrained model, we used Affine BatchNorm. For CIFAR-100, we test on ResNet-32. The hyperparameter choices for each network, dataset and their corresponding sparsities are listed in Tables (9, 10, 11, 12, 13, 14, 15, 16)

### A.3.2 Finetuning the Rare Gems

The details of the hyperparameter we used in finetuning the rare gems we find is shown in Table 17.

## B  Additional Experiments

We repeated the comparison of GEM-MINER with the baselines on MobileNet-V2 on TinyImagenet as well as ResNet-18 on CIFAR-10. We show the results in Fig. 8. Similar to our earlier experiments, we have the following observations. Note that GEM-MINER outperforms IMP (with warmup) in the sparse regime. Also, as is expected, GEM-MINER has non-trivial accuracy before finetune, which is higher than both EP and significantly higher than IMP. GEM-MINER shows a significant deterioration in performance when subjected to the sanity checks suggested in Frankle et al. [11]. Therefore, GEM-MINER is considered to pass the sanity checks. Finally GEM-MINER (*at initialization*) nearly

---

[7]https://pytorch.org/vision/stable/models.html

Table 8: The MobileNet-V2 structure that we use. Each layer consists of 3 total SubnetConv layer which correspond to the respective matrix. Inside the matrix is [kernal×kernal, $C_{out}$]×(number of times to repeat).

| Layer Name | MobileNet-V2 |
|---|---|
| Conv1 | 3×3, 32, stride 1, padding 1 |
| Conv2 | $\begin{bmatrix} 1\times1,\ 32 \\ 3\times3,\ 32 \\ 1\times1,\ 16 \end{bmatrix} \times 1$ |
| Conv3 | $\begin{bmatrix} 1\times1,\ 96 \\ 3\times3,\ 96 \\ 1\times1,\ 24 \end{bmatrix} \times 1$ |
| Conv4 | $\begin{bmatrix} 1\times1,\ 144 \\ 3\times3,\ 144 \\ 1\times1,\ 24 \end{bmatrix} \times 1$ |
| Conv5 | $\begin{bmatrix} 1\times1,\ 144 \\ 3\times3,\ 144 \\ 1\times1,\ 32 \end{bmatrix} \times 1$ |
| Conv6 | $\begin{bmatrix} 1\times1,\ 192 \\ 3\times3,\ 192 \\ 1\times1,\ 32 \end{bmatrix} \times 2$ |
| Conv7 | $\begin{bmatrix} 1\times1,\ 192 \\ 3\times3,\ 192 \\ 1\times1,\ 64 \end{bmatrix} \times 1$ |
| Conv8 | $\begin{bmatrix} 1\times1,\ 384 \\ 3\times3,\ 384 \\ 1\times1,\ 64 \end{bmatrix} \times 3$ |
| Conv9 | $\begin{bmatrix} 1\times1,\ 384 \\ 3\times3,\ 384 \\ 1\times1,\ 96 \end{bmatrix} \times 1$ |
| Conv10 | $\begin{bmatrix} 1\times1,\ 576 \\ 3\times3,\ 576 \\ 1\times1,\ 96 \end{bmatrix} \times 2$ |
| Conv11 | $\begin{bmatrix} 1\times1,\ 576 \\ 3\times3,\ 576 \\ 1\times1,\ 160 \end{bmatrix} \times 1$ |
| Conv12 | $\begin{bmatrix} 1\times1,\ 960 \\ 3\times3,\ 960 \\ 1\times1,\ 160 \end{bmatrix} \times 2$ |
| Conv13 | $\begin{bmatrix} 1\times1,\ 960 \\ 3\times3,\ 960 \\ 1\times1,\ 320 \end{bmatrix} \times 1$ |
| FC1 | 320×1280 |
| FC2 | 1280×10 |

Table 9: Hyper Parameters used for different sparsities for GEM-MINER on ResNet-20 on CIFAR-10.

| Network/Dataset | Sparsity | Pruning Period | Optimizer | LR | Lambda |
|---|---|---|---|---|---|
| ResNet-20 CIFAR-10 | 50% | 8 | SGD | 0.05 | $10^{-8}$ |
| | 86.26% | 5 | SGD | 0.1 | $10^{-5}$ |
| | 96.27% | 5 | SGD | 0.1 | $3 \times 10^{-5}$ |
| | 98.56% | 5 | SGD | 0.1 | $10^{-4}$ |
| | 99.41% | 5 | SGD | 0.1 | $10^{-4}$ |

Table 10: Hyper Parameters used for different sparsities for GEM-MINER on MobileNet-V2 on CIFAR-10.

| Network/Dataset | Sparsity | Pruning Period | Optimizer | LR | Lambda |
|---|---|---|---|---|---|
| MobileNet-V2 CIFAR-10 | 50% | 5 | SGD | 0.05 | 0 |
| | 80% | 5 | SGD | 0.1 | $3 \times 10^{-6}$ |
| | 95% | 5 | SGD | 0.1 | $7 \times 10^{-6}$ |
| | 98.56% | 5 | SGD | 0.1 | $10^{-4}$ |

Table 11: Hyper Parameters used for different sparsities for GEM-MINER on VGG-16 on CIFAR-10.

| Network/Dataset | Sparsity | Pruning Period | Optimizer | LR | Lambda |
|---|---|---|---|---|---|
| VGG-16 CIFAR-10 | 50% | 5 | SGD | 0.01 | 0 |
| | 95% | 5 | SGD | 0.01 | $10^{-6}$ |
| | 97.5% | 5 | SGD | 0.01 | $10^{-6}$ |
| | 98.6% | 5 | SGD | 0.01 | $10^{-6}$ |
| | 99.5% | 5 | SGD | 0.01 | $10^{-6}$ |

Table 12: Hyper Parameters used for different sparsities for GEM-MINER on WideResNet-28-2 on CIFAR-10.

| Network/Dataset | Sparsity | Pruning Period | Optimizer | LR | Lambda |
|---|---|---|---|---|---|
| WideResNet-28-2 CIFAR-10 | 50% | 5 | SGD | 0.1 | 0 |
| | 80% | 10 | SGD | 0.1 | $10^{-5}$ |
| | 95% | 10 | SGD | 0.1 | $10^{-5}$ |
| | 98.56% | 10 | SGD | 0.1 | $10^{-5}$ |
| | 99.5% | 10 | SGD | 0.1 | $10^{-5}$ |

Table 13: Hyper Parameters used for different sparsities for GEM-MINER on ResNet-18 on TinyImageNet.

| Network/Dataset | Sparsity | Pruning Period | Optimizer | LR | Lambda |
|---|---|---|---|---|---|
| ResNet-18 TinyImageNet | 50% | 10 | SGD | 0.1 | 0 |
| | 95% | 5 | SGD | 0.001 | $8 \times 10^{-6}$ |
| | 98.6% | 5 | SGD | 0.001 | $5 \times 10^{-6}$ |
| | 99.5% | 5 | SGD | 0.001 | $10^{-5}$ |

Table 14: Hyper Parameters used for different sparsities for GEM-MINER on ResNet-50 on TinyImageNet.

| Network/Dataset | Sparsity | Pruning Period | Optimizer | LR | Lambda |
|---|---|---|---|---|---|
| ResNet-50 TinyImageNet | 50% | 5 | ADAM | 0.01 | 0 |
| | 95% | 5 | ADAM | 0.01 | $10^{-6}$ |
| | 98.6% | 5 | ADAM | 0.01 | $10^{-6}$ |
| | 99.5% | 5 | ADAM | 0.01 | $10^{-6}$ |

Table 15: Hyper Parameters used for different sparsities for GEM-MINER on ResNet-50 on Caltech101.

| Network/Dataset | Sparsity | Pruning Period | Optimizer | LR | Lambda |
|---|---|---|---|---|---|
| ResNet-50 Caltech101 | 50% | 5 | ADAM | 0.01 | 0 |
| | 95% | 5 | ADAM | 0.01 | $10^{-6}$ |
| | 98.6% | 5 | ADAM | 0.01 | $10^{-6}$ |
| | 99.5% | 5 | ADAM | 0.01 | $10^{-6}$ |

achieves the performance of **Renda et al.** (Learning rate rewinding) in the sparse regime. This is particularly impressive given that **Renda et al.** is a pruning-after-training method *i.e.*, it prunes and trains iteratively, never finding a subnetwork at or even near initiailization.

Table 16: Hyper Parameters used for different sparsities for GEM-MINER on ResNet-32 on CIFAR-100.

| Network/Dataset | Sparsity | Pruning Period | Optimizer | LR | Lambda |
|---|---|---|---|---|---|
| ResNet-32 CIFAR-100 | 50% | 5 | SGD | 0.1 | $10^{-7}$ |
| | 80% | 5 | SGD | 0.1 | $10^{-6}$ |
| | 95% | 5 | SGD | 0.1 | $10^{-5}$ |
| | 98% | 5 | SGD | 0.1 | $10^{-5}$ |
| | 99.5% | 5 | SGD | 0.1 | $5 \times 10^{-5}$ |

Table 17: Hyperparameters used for finetuning. We use the same number of epochs for GEM-MINER and finetuning.

| Model | Dataset | Sparsity | Epochs | Batch Size | LR | LR-Schedule Milestones |
|---|---|---|---|---|---|---|
| ResNet-20 | CIFAR-10 | 50% | 150 | 128 | 0.1 | [80, 120] |
| | CIFAR-10 | others | 150 | 128 | 0.01 | [80, 120] |
| MobileNet-V2 | CIFAR-10 | all | 300 | 128 | 0.1 | [150, 250] |
| VGG-16 | CIFAR-10 | all | 200 | 128 | 0.05 | [100, 150] |
| WideResNet-28-2 | CIFAR-10 | 50% | 150 | 128 | 0.1 | [80, 120] |
| | CIFAR-10 | others | 150 | 128 | 0.01 | [80, 120] |
| ResNet-18 | TinyImageNet | all | 200 | 256 | 0.1 | [100, 150] |
| ResNet-50 | TinyImageNet | all | 150 | 256 | 0.1 | [80, 120] |
| ResNet-50 (Pretrained) | Caltech-101 | all | 50 | 16 | 0.0001 | [20, 40] |
| ResNet-32 | CIFAR-100 | all | 150 | 64 | 0.01 | [80, 120] |

**How far can we take random pruning?** Recall that subnetworks identified by GEM-MINER outperform random subnetworks found by sampling based on smart ratio (SR) [41]. However, SR performs surprisingly well given that it is still random pruning. Therefore, we tried improving SR to see how far we could take random pruning. For simplicity, we call these different versions of SR starting with **v1** which is the original algorithm suggested by Su et al. [41]. Table 18 compares the post-finetune accuracy of subnetworks found by training randomly initialized subnetworks at initialization, each using "smart ratios" found by increasingly sophisticated means. Given $L$-layer network, each version of SR uses its own sparsity pattern $\boldsymbol{p} = [p_1, \cdots, p_L]$ to generate random subnetwork, where each layer randomly picks $0 < p_l \leq 1$ fraction of weights to use. The sparsity patterns are identified as follows:

- *SR-v1*: vanilla SR suggested in [41]
- *SR-v2*: set $p_l = p_l^{\text{SR}}$ for $2 \leq l \leq L - 1$ and $p_l = p_l^{\text{GM}}$ for $l \in \{1, L\}$
- *SR-v3*: set $p_l = p_l^{\text{SR}}$ for $2 \leq l \leq L - 1$ and $p_l = 1$ for $l \in \{1, L\}$
- *SR-v4*: start with $\boldsymbol{p}^{\text{IMP}}$ and search sparsity patterns in a small ball around it.
- *SR-v5*: start with the sparsity pattern of v2, and tune $p_l$ using Eq (1)
- *SR-v6*: start with the sparsity pattern of v4, and tune $p_l$ using Eq (1)

The vanilla SR proposed by Su et al. [41] is denoted by *SR-v1*. Motivated by the fact that (i) GEM-MINER outperforms SR and (ii) GEM-MINER and SR primarily differ in the layerwise sparsity $p_l$ for the first and the last layers (refer Figure 7), we construct *SR-v2*. It uses the layerwise sparsity of *SR-v1* for intermediate layers ($2 \leq l \leq L$) and layerwise sparsity of GEM-MINER for $l \in \{1, L\}$. As a simple variant of *SR-v2*, we also considered using full dense layer ($p_l = 1$) for the first and the last layer, which is denoted by *SR-v3*. This was because we observed that the first and last layers found by GEM-MINER were relatively dense compared to *SR-v1*. *SR-v4* searches a few points around the sparsity pattern of IMP and chooses the best option. We chose only a few options based on intuition to make the search computationally tractable.

Finally, we tried to formulate finding the optimal random pruning method by writing it as an optimization problem in $p_l$ as follows:

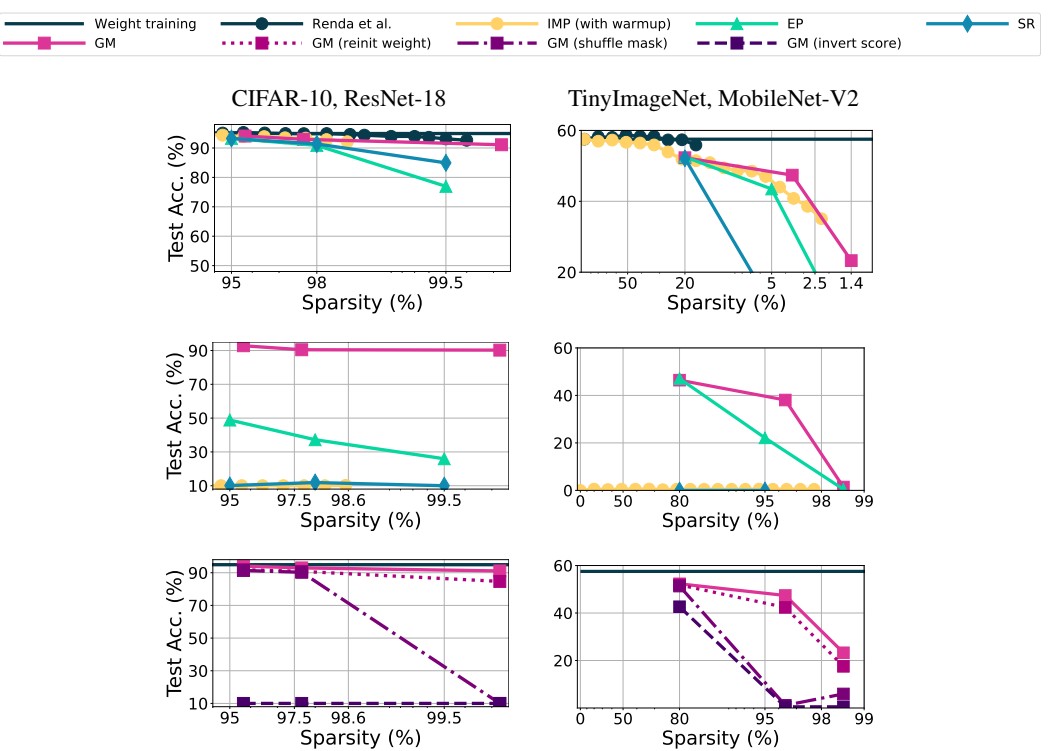

Figure 8: Additional experimental results on comparing different pruning algorithms. Top: post-finetune accuracy, Middle: pre-finetune accuracy, Bottom: sanity check methods suggested in Frankle et al. [11] applied on GEM-MINER. Similar to earlier experiments, we find that GEM-MINER outperforms IMP (with warmup) in the sparse regime in terms of both post finetune as well as pre-finetune accuracy. We also note that GEM-MINER passes all of the sanity checks.

Table 18: Performance comparison of GEM-MINER and variants of Smart Ratio (SR), for ResNet-20 trained for CIFAR-10 classification task, for target sparsity 98.56%. We denoted the vanilla SR in [41] as *SR-v1* and tested five additional variants, from *SR-v2* to *SR-v6*. The detailed description of the variants are given in Paragraph B.

| Schemes | Gem-Miner | SR (v1) [41] | SR (v2) | SR (v3) | SR (v4) | SR (v5) | SR (v6) |
|---|---|---|---|---|---|---|---|
| Sparsity (%) | 98.56 | 98.56 | 98.53 | 98.25 | 98.56 | 98.47 | 98.53 |
| Accuracy (%) | **77.89** | 65.61 | 68.59 | 69.78 | 69.92 | 69.01 | 69.08 |

$$\min_{p_l} \sum_{(\boldsymbol{x},y) \in S} \ell(f(\boldsymbol{w}_l \odot \text{Bern}(p_l); \boldsymbol{x}), y) \tag{1}$$

Intuitively, this is equivalent to choosing $p_l$ such that the loss of the random subnetwork generated by sampling the mask of layer $l$ with probability $p_l$ is minimized.

In order to make the problem more tractable, we set the output of previous versions as the initial value of $p_l$. Choosing $p_l^{(0)}$ as the $p_l^{\text{SR-v2}}$ and then applying SGD on Eq 1 gives us *SR-v5*. Repeating this with $p_l^{(0)} = p_l^{\text{SR-v4}}$ results in *SR-v6*.

The results of Table 18 show that different strategies in random pruning can improve the performance of random pruning, but GEM-MINER still has an 8% accuracy gap with the best random network we found for ResNet-20, CIFAR-10 classification, when the target sparsity is 98.56%.

**Relationship between pre-finetune accuracy and post-finetune accuracy.** Recall that rare gems need to have not only high *post-finetune* accuracy but also non-trivial *pre-finetune* accuracy.

Since the latter is a lower bound on *post-finetune* accuracy, we design GEM-MINER to just maximize the accuracy at initialization. However, it is not clear that this actually maximizes post-finetune accuracy. In fact, the performance of EP and IMP clearly show that it is neither a necessary nor sufficient condition. Fig. 9 shows both pre and post finetune accuracies for GEM-MINER (GM), edge-popup (EP), and smart ratio (SR), at both 98.6% and 99.5% sparsity, for CIFAR-10 classification using VGG-16. For GEM-MINER, we show the results of using four different learning rates $\eta = 0.01, 0.001, 0.0001, 0.00001$ for 200 epochs using lighter colors to indicate lower learning rates. It turns out that $\eta = 0.01$ has the best pre and post-finetune accuracies for both sparsities. Moreover, it shows that there is some correlation between pre-finetune and post-finetune accuracy *i.e.*, subnetworks that have higher pre-finetune

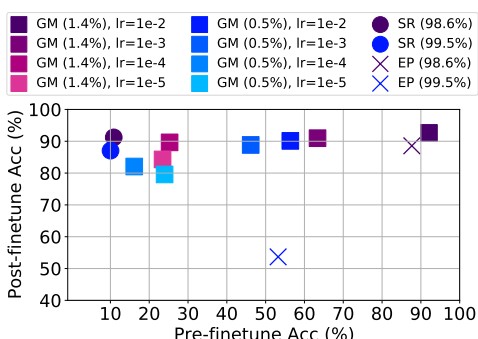

Figure 9: Relationship between pre-finetune accuracy and post-finetune accuracy, on CIFAR-10, VGG-16, for GEM-MINER (GM), edge-popup (EP), and smart ratio (SR). For both 98.6% and 99.5% sparsity, GEM-MINER typically shows higher post-finetune accuracy when it has higher pre-finetune acuracy. As noted by Ramanujan et al. [33], subnetworks found by EP are not finetunable.

accuracies typically have higher accuracy after finetuning as well. However, comparing GEM-MINER's results for 99.5% sparsity with $\eta = 0.0001$ and $\eta = 0.00001$ shows that this pattern does not always hold. With $\eta = 0.00001$, GEM-MINER achieves a higher pre-finetune accuracy but ends up with lower accuracy after finetuning. Therefore, we conclude that while pre-finetune accuracy is a reasonable proxy for accuracy after finetuning, it does not guarantee it in any way. Note that both points for EP have "post-finetune accuracy" $\simeq$ "pre-finetune accuracy", which confirms the observation by Ramanujan et al. [33] that subnetworks found by EP are not lottery tickets *i.e.*, they are not conducive to further training.

**Discussions on the need of warmup for IMP.** For completeness, we also tried different variants of IMP. We refer to it as "cold" IMP when the weights are rewound to initialization, while "warm" IMP rewinds to some early iteration, *i.e.*, after training for a few epochs. Further, we classify it depending on the number of epochs per magnitude pruning. We say IMP is "short" if it only trains for a few epochs (*e.g.*, 8 epochs for ResNet-20 on CIFAR-10) before pruning, and "long" if it takes a considerably larger number of epochs before pruning (*e.g.*, 160 epochs for ResNet-20 on CIFAR-10). Regardless of "long" or "short", the number of epochs to finetune the pruned model are the same.

With these informal definitions, we can categorize IMP into four different versions: a) *short-cold* IMP, b) *short-warm* IMP, c) *long-cold* IMP, and d) *long-warm* IMP.

In the literature, only *long* variants have been studied thoroughly. Frankle et al. [10] noted that for large networks and difficult tasks, long-cold IMP fails to find lottery tickets, which is why they introduce long-warm IMP.

Somewhat surprisingly, as shown in Fig. 10, we find that short versions of IMP can achieve the same or an even better accuracy-sparsity trade-off especially in the sparse regime. In particular, *short-cold* IMP matches the performance of *long-warm* one without any warmup, *i.e.*, *short-cold* IMP can find lottery-tickets at initialization. However, note that *short-cold* IMP only finds lottery tickets, **not** rare gems in that the subnetworks it finds have accuracy close to that of random guessing.

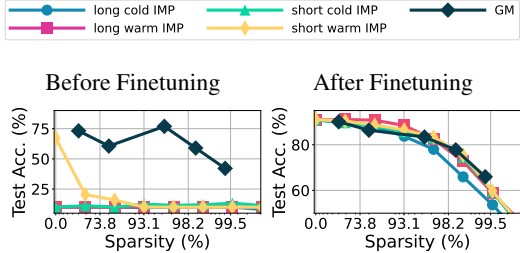

Figure 10: Test accuracy before (*left*) and after (*right*) finetuning for ResNet-20 on CIFAR-10. The "short" version of IMP achieves the same accuracy-sparsity tradeoff as *long-warm* IMP in the sparse regime. However, comparing the before finetuning accuracy (*left*) shows that GEM-MINER is capable of finding rare gems at initialization, whereas *short-cold* IMP can only find lottery tickets.