# OpenReview forum: "Rare Gems: Finding Lottery Tickets at Initialization"
_NeurIPS.cc/2022/Conference — NeurIPS 2022 Accept_

### Official Review · Reviewer_YCDn · 2022-07-04

**Rating:** 6
**Confidence:** 4
**Soundness:** 3 good
**Presentation:** 3 good
**Contribution:** 2 fair

**Summary:**

This paper studies the problem of searching trainable subnetworks at initialization (winning tickets). The authors propose a method called GEM-MINER, which maximises the accuracy of subnetworks before weight training. GEM-MINER can discover subnetworks comparable to IMP with warmup (a widely-used method to find winning ticket subnetworks) while being 19 $\times$ faster. GEM-MINER outperforms the Edge-popup, which is another method that also optimizes the subnetwork structure before weight training. The subnetworks found by GEM-MINER also pass the "sanity check", outperforming random sparse subnetworks.

**Questions:**

* Why not compare with SNIP, GraSP and SynFlow?
* Could you remark on how to resolve the limitations you discussed in line 311-321?

**Limitations:**

Please refer to the Strengths And Weaknesses and Questions.

**Strengths And Weaknesses:**

### Strengths
* Searching for subnetworks at initialization with better after-training performance is an important problem in the lottery ticket hypothesis (LTH). This paper proposes a new method to tackle this problem and achieves promising results.
* The main reasons (i.e., global pruning and gradual pruning) for the effectiveness of the proposed method are well explained in the ablation study (Tab.3), which can be used as default setups in follow-up works.
* Extensive experiments are conducted to demonstrate the effectiveness of the proposal, covering multiple tasks and backbone models.
* The paper is generally well-written and easy to follow.

### Weaknesses
1. Although the proposal is a new method to search trainable subnetworks at initialization, its core ideas (global pruning and gradual pruning) have already been studied in the literature on model compression [1]. This undermines the technical novelty of this paper.
[1] Movement Pruning: Adaptive Sparsity by Fine-Tuning. In NeurIPS 2020.

2. The effectiveness of global pruning and gradual pruning is a useful finding. However, it is unclear how these two factors contribute to identifying more trainable subnetworks at initialization. Further analysis is required.

3.  Although extensive experiments are conducted, the empirical studies can still be improved in terms of comprehensiveness. For example:
* Comparing with other "pruning at initialization" methods (e.g., SNIP, GraSP and SynFlow).
* The paragraph "Convergence of accuracy and sparsity" can benefit from adding results of Edge-popup and SNIP, etc.

4. Some arguments made in this paper are questionable. For example, "EP subnetworks are not conducive to further fine-tuning" does not always hold. At 5% sparsity, the accuracy of EP subnetworks can be improved from 70.69 to 84.34, according to Fig.2.

5. Limitations of the GEM-MINER are well discussed. However, the authors did not provide any ways of solutions.

### Typos
* In the upper line of Fig.1 (the pipeline of IMP), it seems that the "0.5% sparse Subnetwork at Warm Initialization" should have the same color as the "Partially trained dense network" for the same weights.
* CIFAR-100 is not mentioned in the paragraph "Tasks" (line 226-230), while appears in Fig.4.
* The caption of Fig.4 mentions ImageNet, while only TinyImageNet can be found in the titles of 2nd and 3rd columns.
* Line 272 mentions Fig.4c, which cannot be found in Fig.4.

---

> ### Author Response · Authors · 2022-08-02
> **Response to Reviewer YCDn (Part 1)**
>
> We are glad that the reviewer also feels that the problem of pruning at initialization is important in the LTH literature. We are excited that they found the analysis useful and also hope that it can help guide future research in the area.
>
> `Global pruning and gradual pruning have been studied in Model Compression`
> > Although the proposal is a new method to search trainable subnetworks at initialization, its core ideas (global pruning and gradual pruning) have already been studied in the literature on model compression [1]. This undermines the technical novelty of this paper. [1] Movement Pruning: Adaptive Sparsity by Fine-Tuning. In NeurIPS 2020.
>
> We would like to emphasize that **our claim to novelty is not the algorithm (Gem-Miner), but the discovery of Rare Gems (per our title and abstract).** We establish the existence of lottery tickets in popular settings (CIFAR10/tinyImageNet, and others); these are setting where other algorithms fail sanity checks.
>
> We agree that several of these techniques have been applied in model compression and Movement pruning[1] specifically utilizes the idea of computing the mask as a quantized version of the scores. However, they study it in the setting of training and pruning simultaneously and not for identifying subnetworks at initialization. This should not cast any negative light on what Gem-Miner discovers: trainable subnetworks at initialization, in popular settings, where other algorithms failed. And we feel that our work most certainly does not diminish the technical novelty of prior model compression algorithms.
>
> We have amended the manuscript to include a discussion on [1] under related work. That said, it would be interesting to see if these techniques alone are sufficient to reach the same performance. We do this using an ablation study where we consider each of these ideas independently. We find that in fact, regularization is the most important tool in addition to quantizing the scores. However, as we discuss in Section 3 of the paper, regularization alone is insufficient to exhibit fine-grained control over the target sparsity. Removing regularization and only using gradual pruning, however, leads to a massive drop in accuracy. Further removing global pruning and using just "_local_" iterative freezing ends up performing scarcely better than random.
>
> **[Table R1] Ablation study of Gem-Miner on ResNet20, CIFAR-10, Sparsity 1.44%**
>
> |                GM-Variant                 | Acc before FT | Acc after FT  |
> |:-----------------------------------------:|:----------------------------:|:---------------------------:|
> |                Gem-Miner (proposed)                 |         $61.23\%$          |         $77.12\%$         |
> |    Gem-Miner - Regularization    |         $10.18\%$          |         $27.41\%$         |
> | Gem-Miner - Regularization - Global pruning |         $10.08\%$          |         $11.6\%$         |
> |||
>
> As we can see from Table R1, Gem-Miner outperforms all of its ablated versions. More precisely, Gem-Miner without regularization works extremely poorly, and Gem-Miner without regularization and global pruning is barely above random guessing. Note that when we ablate global pruning, the algorithm chooses the bottom-$k$ weights in each layer much like Edge-Popup.
>
> `Further analysis on why global pruning and gradual pruning works`
> > The effectiveness of global pruning and gradual pruning is a useful finding. However, it is unclear how these two factors contribute to identifying more trainable subnetworks at initialization. Further analysis is required.
>
> We agree that it would be nice to better understand what exactly contributes to the "_trainability_" of these subnetworks. We believe that Fig 7 in our paper gives us some hints on the necessity of global pruning by comparing the layerwise sparsities of the subnetworks identified by different algorithms. For example, it seems plausible that the most effective way to distribute the "_sparsity budget_" is to spend it on the first and last layers.
>
> We find empirically that gradual pruning may not even be necessary if the regularization parameter is chosen extremely carefully. However, as we remark in Section 3, finding the appropriate regularization weight is quite challenging, especially to achieve a specific target sparsity. Thus, we find that gradual pruning is useful to ensure that the final subnetwork is of the target sparsity.
>
> We will include this interpretation in the final manuscript.

---

> > ### Author Response · Authors · 2022-08-02
> > **Response to Reviewer YCDn (Part 2)**
> >
> > `Empirical studies can still be improved in terms of comprehensiveness.`
> > > Although extensive experiments are conducted, the empirical studies can still be improved in terms of comprehensiveness. For example: Comparing with other "pruning at initialization" methods (e.g., SNIP, GraSP and SynFlow). The paragraph "Convergence of accuracy and sparsity" can benefit from adding results of Edge-popup and SNIP, etc.
> >
> > > Why not compare with SNIP, GraSP and SynFlow?
> >
> > We thank the reviewer for acknowledging that our experiments are extensive. With regards to SNIP, GraSP and SynFlow, we found that since Frankle et al. already showed that these pruning at initialization methods fail to pass the sanity checks, it was unnecessary to repeat those experiments. However, we will try to include these baselines in the final manuscript for a few settings. We would like to add that ProsPr (Alizadeh et al.) is also a pruning at init method which was published more recently. Therefore, we included a comparison with it in Section 4.2.
> >
> >
> > `Including Edge-popup for convergence.`
> > > The paragraph “Convergence of accuracy and sparsity” can benefit from adding results of Edge-popup and SNIP, etc.
> >
> > We have updated the plot to include Edge-popup. We will try to include a few more baselines in the final manuscript.
> >
> > `EP subnetworks are not conducive to further fine-tuning`
> > > Some arguments made in this paper are questionable. For example, "EP subnetworks are not conducive to further fine-tuning" does not always hold. At 5% sparsity, the accuracy of EP subnetworks can be improved from 70.69 to 84.34, according to Fig.2.
> >
> > We would first like to point out that this is in fact a claim made by the original authors of EP in their paper by Ramanujan et al. However, you are right in that it does not always hold and the term "_conducive to finetuning_" is somewhat imprecise. We will clarify this in the final manuscript.
> >
> >
> > `Limitations are well discussed, but what about solutions?`
> > > Limitations of the GEM-MINER are well discussed. However, the authors did not provide any ways of solutions.
> >
> > > Could you remark on how to resolve the limitations you discussed in line 311-321?
> >
> > We include the limitation section as our attempt, to be honest, and forthcoming with regards to what can be improved with respect to the current shortcomings of our algorithm. Due to space limitations, we focused on giving high-level ideas of the main concerns, with some ideas on how to improve them. If those were missing, it is mostly because we have not yet figured out how to resolve them in a concrete manner, although we have some ideas on how to move forward. Since the openreview discussion is open to the public, we hope that the interested reader will also spend time reading these to get a more clear picture, beyond the main text. We will also spend more time in the appendix to go over these issues.
> >
> > - ``[Gap with EP]`` As we discuss in the paper, more careful tuning of hyperparameters is likely to resolve the gap with EP.
> > - `[Layer-collapse]` We find that reasonable choices of $\lambda$ avoid layer-collapse quite easily but we state it as a possibility when a practitioner tries Gem-Miner using different hyperparameter choices.
> > - `[Sensitivity to hyperparameters]` This seems to be an artifact of the hardness of the problem as well as the algorithm itself. It is conceivable that making some of these parameters learnable could solve this issue but we have not yet been able to fix this completely.
> >
> > `Typos`
> > > In the upper line of Fig.1 (the pipeline of IMP), it seems that the “0.5% sparse Subnetwork at Warm Initialization” should have the same color as the “Partially trained dense network” for the same weights.
> >
> > Thank you for your extremely careful reading! You are right. We will fix it.
> >
> > > CIFAR-100 is not mentioned in the paragraph “Tasks” (line 226-230), while appears in Fig.4.
> >
> > Thank you for pointing it out, we seem to have missed it. We will include it.
> >
> > > The caption of Fig.4 mentions ImageNet, while only TinyImageNet can be found in the titles of 2nd and 3rd columns.
> >
> > Thank you for your careful reading. It should be CIFAR-100. We have changed it.

---

> > ### Comment · Reviewer_YCDn · 2022-08-09
> > **Response**
> >
> > Thank you for taking the time to respond to my comments. My concerns regarding weaknesses 1,3,4,5 are generally addressed. As for the second weakness, I think the authors' response still cannot explain why global pruning and gradual pruning are useful. For example, the discussion about Fig.7 is still a phenomenon, not a reason. Nevertheless, I believe the findings of this work can facilitate related research on finding trainable subnetworks at initialization and thus I will retain my rating.

---

### Official Review · Reviewer_Loxt · 2022-07-11

**Rating:** 8
**Confidence:** 4
**Soundness:** 4 excellent
**Presentation:** 4 excellent
**Contribution:** 4 excellent

**Summary:**

The paper proposes GEM-MINER to find lottery tickets at initialization. The found subnetworks are trainable and can achieve comparable or better performance compared to existing approaches (e.g., IMP), and requires much less time (19x faster compared to IMP). The approach can pass the sanity checks proposed by prior work. The experiments with different datasets, different model architectures, and different sparisities show the effectiveness of the proposed approach. The paper claims it resolves the open problem of hunting for subnetworks at initialization trainable to SOTA accuracy.

**Questions:**

- Can GEM-MINER be applied to NLP tasks?

**Limitations:**

The limitations have been well discussed in the paper in an explicit section (e.g., it may require careful hyper-parameters tuning). I don’t see other major limitations of the approach.

**Strengths And Weaknesses:**

Strengths:
- The paper is well-written and very easy to understand.
- The paper conducts extensive and solid experiments to study the effectiveness of the proposed method. The results are strong and exciting, and also provide insightful analysis (e.g., why EP failed).
- The approach is simple and effective. I believe that the paper can facilitate more research in this field.

Weaknesses:
I don’t see a major concern with this paper.

Minor: why is sparisity defined as the ratio of non-zero weights instead of zero weights? (Based on the current definition, 95% sparsity actually means a very dense model!)

---

> ### Author Response · Authors · 2022-08-02
> **Response to Reviewer Loxt**
>
> We are encouraged that the reviewer finds that our results are strong and exciting and finds our analysis insightful. We too hope that it will facilitate future research in the field.
>
> `Definition of sparsity`
> > Minor: why is sparisity defined as the ratio of non-zero weights instead of zero weights? (Based on the current definition, 95% sparsity actually means a very dense model!)
>
> Thank you for pointing it out. We will change it in the final version.
>
> `Extension to NLP tasks`
> > Can GEM-MINER be applied to NLP tasks?
>
> This is an excellent suggestion. We believe that since several other pruning algorithms have shown success on NLP tasks (Chen et al., Prasanna et al.), Gem-Miner too should work in that setting. However, since most NLP tasks are for fine-tuning as opposed to training from random initialization, we decided to start with vision tasks.
> We are running some experiments and hope to include the results in the final version.
>
> (**References:** Chen et al., _"The Lottery Ticket Hypothesis for Pre-trained BERT Networks"_, Prasanna et al., _"When BERT Plays the Lottery, All Tickets Are Winning"_)

---

### Official Review · Reviewer_nnnE · 2022-07-12

**Rating:** 5
**Confidence:** 3
**Soundness:** 2 fair
**Presentation:** 2 fair
**Contribution:** 3 good

**Summary:**

This work aims to find lottery tickets at initialization efficiently by a proposed Gem-Miner.
Previous studies either find bad tickets at initialization (ineffectiveness) or find good tickets after initialization (inefficiency).
To tackle the open problem, the Gem-Miner is proposed to discover rare gems, which can find good enough tickets at initialization even before weight training.
The Gem-Miner can be essentially viewed as a bag of tricks for finding tickets.
Experimental results show that rare gems perform far better than baselines in terms of both performance metrics and sanity checks.
In addition, rare gems hold the advantage of a faster identification process.


**Questions:**

Q1:
The authors state that "exactly the same number of epochs" yet "faster than IMP" at the same time in Line 61, which seems ambiguous to me. Would "IMP" consume longer training time for an epoch?
If I understand correctly, the "IMP" in Table 1 indicates "IMP with warmup" instead since it passes sanity checks as stated in Line 40. Then it can be acknowledged that Gem-Miner is faster than "IMP with warmup". However, this would be controversial to the above statement.

Q2:
"after E epochs of weight training" in Line 232 indicates that a pre-training stage is required, so I wonder whether other pruning methods require a pre-training stage if the pre-training is required by Edge-Popup. Would this lead to an unfair comparison?

**Limitations:**

Hyper-parameters are sensitive in Gem-Miner but are not listed in this paper.



**Strengths And Weaknesses:**

Pros:
1. The open problem is interesting and is somewhat solved by the proposed method.
2. Appealing experiments results in terms of both effectiveness and efficiency.

Cons:
1. There are several presentation flaws as listed in the Questions below.
2. This work focuses on a specific pruning method (i.e., Edge-Popup), and I would glad to see further results of arming Gem-Miner to other pruning methods.
3. This work can be viewed as a bag of tricks for finding good tickets at initialization. However, ablation studies are not thoroughly conducted for a better understanding. There is a related section describing how Gem-Miner resolves Edge-Popups failings by adding components to Edge-Popup, yet without adequate connections between these components and tricks introduced by Gem-Miner.

---

> ### Author Response · Authors · 2022-08-02
> **Response to Reviewer nnnE**
>
> We are glad that the reviewer also believes the open problem is interesting and that our experimental results are appealing.
>
> **`Ablation study and "arming" Gem-Miner to other algorithms`**
> > This work focuses on a specific pruning method (i.e., Edge-Popup), and I would glad to see further results of arming Gem-Miner to other pruning methods.
> This work can be viewed as a bag of tricks for finding good tickets at initialization. However, ablation studies are not thoroughly conducted for a better understanding. There is a related section describing how Gem-Miner resolves Edge-Popups failings by adding components to Edge-Popup, yet without adequate connections between these components and tricks introduced by Gem-Miner.
>
> **Ablation study**
>
> As suggested by the reviewer, we run the following ablation study to better understand the relative importance of the different components of the algorithm. Here are the results:
>
> **[Table R1] Ablation study of Gem-Miner on ResNet20, CIFAR-10, Sparsity 1.44%**
>
> |                GM-Variant                 | Acc before FT | Acc after FT  |
> |:-----------------------------------------:|:----------------------------:|:---------------------------:|
> |                Gem-Miner (proposed)                 |         $61.23\%$          |         $77.12\%$         |
> |    Gem-Miner - Regularization    |         $10.18\%$          |         $27.41\%$         |
> | Gem-Miner - Regularization - Global pruning |         $10.08\%$          |         $11.6\%$         |
> |||
>
> As we can see from Table R1, Gem-Miner outperforms all of its ablated versions. More precisely, Gem-Miner without regularization works extremely poorly, and Gem-Miner without regularization and global pruning is barely above random guessing. Note that when we ablate global pruning, the algorithm chooses the bottom-$k$ weights in each layer much like Edge-Popup.
>
> **Arming Gem-Miner to other pruning algorithms**
>
> We would also like to clarify that it is not always possible to "arm" Gem-Miner to different pruning algorithms. For example, it is not clear how to apply regularization say, to SNIP or some other single-shot pruning method since the score for each weight is computed as a function of the weights and not learned through optimization. Similarly, including a quantization function over the scores works only in situations where scores are assigned to each weight and then learned by backpropagating over the network. However, we acknowledge that it is an interesting direction to see if there exist other pruning algorithms that can benefit from adding some of the components of Gem-Miner. We hope to explore it going forward.
>
>
> `Clarification Q1: Exactly the same number of epochs yet faster than IMP`
> > Q1: The authors state that “exactly the same number of epochs” yet “faster than IMP” at the same time in Line 61, which seems ambiguous to me. Would “IMP” consume longer training time for an epoch? If I understand correctly, the “IMP” in Table 1 indicates “IMP with warmup” instead since it passes sanity checks as stated in Line 40. Then it can be acknowledged that Gem-Miner is faster than “IMP with warmup”. However, this would be controversial to the above statement.
>
> We apologize for the confusion. We merely state that Gem-Miner takes the same number of epochs to find the subnetwork as it takes to then finetune the subnetwork to full accuracy. We have updated the manuscript to make the language more clear.
> The time taken for both Gem-Miner and IMP over an epoch is quite similar since they both require doing backprop over the network. However, IMP is iterative and repeats this process for several _"rounds"_ to reach a sufficiently sparse model. This is why it takes $19\times$ more time. Therefore, there is no contradiction.
>
> `Clarification Q2: Pre-training is required?`
> > Q2: “after E epochs of weight training” in Line 232 indicates that a pre-training stage is required, so I wonder whether other pruning methods require a pre-training stage if the pre-training is required by Edge-Popup. Would this lead to an unfair comparison?
>
> Thank you for your careful reading. We would like to emphasize that there is no pretraining required for both Edge-Popup (EP) and Gem-Miner. The most important feature of Gem-Miner is that it finds subnetworks _at initialization_ without the need for warmup. We apologize for the confusion. We just wanted to specify how many epochs Gem-Miner needs (typically), in order to find a sparse, accurate subnetwork. If dense weight training of a randomly-initialized network takes $E$ epochs, then we find that is sufficient to run Gem-Miner for $E$ epochs on the same randomly initialized network for it to find a sparse, accurate subnetwork. In fact, this is usually the case for EP as well.
>
> `Clarification Q3: Hyper-parameters are sensitive in Gem-Miner but are not listed in this paper.`
>
> We would like to refer the reviewer to the tables in Appendix A.3 where we list all of the hyperparameter choices in detail.

---

> > ### Comment · Reviewer_nnnE · 2022-08-08
> > **Feedback to the Authors**
> >
> > Thanks for your responses! The clarifications have addressed my questions. The ablation study is also well-designed.
> >
> > However, I am still concerned that Gem-Miner is a bag of tricks designed specifically upon Edge-Popup so that rare gems can be discovered. Meanwhile, the reasons why these tricks can enable Edge-Popup to discover rare gems remain unclear to me.
> >
> > So I would like to keep my current rating.

---

> > > ### Author Response · Authors · 2022-08-08
> > > **Response to Reviewer nnnE comment**
> > >
> > > We would like to thank the Reviewer for engaging with us during this phase, and we hope we can address the issues raised. It would be very helpful if the reviewer could suggest further ablation studies that would better explain why Gem-Miner works beyond the ones we already conducted.
> > >
> > > Gem-Miner is attempting to address the shortcomings of edge-popup by using some principled ideas: $[0,1]$ bounded mask scores, regularization, and weight thresholding. We view this as a bag of tricks in the same way that any training algorithm used in practice is a bag of tricks. We do not claim to reinvent any of the algorithmic ideas used, but that we make the right choice in the algorithmic principles we put together. Since the hunt for lottery tickets has been long and yielded no sanity-check passing algorithms, we believe that the approach of "_finding the right ideas that work_" should not cast a negative light on our main claim to novelty: finding lottery tickets _at initialization_ for several tasks which was an open problem.
> > >
> > > The reason why we build on Edge-Popup is precisely because starting with a high accuracy network *at* initialization seems like a reasonable starting step to get high accuracy after fine-tuning with SGD. As far as we know, none of the other algorithms in the related literature attempts to find high accuracy networks at initialization apart from EP. Yet EP's shortcoming is that the networks it finds are not significantly fine-tunable. Fixing that shortcoming seemed like a reasonable goal, and indeed by taking the principled approaches listed above, we were able to address its issues. The fact that several papers after the original work of Frankle and Carbin, and the paper by Ramanujan et al. did not manage to find an algorithm that passes the sanity checks provided by the community is what we think is of value in our contribution.
> > >
> > > That said, we would appreciate it if the reviewer clarifies their concerns so we can improve our paper according to their feedback. Is the lack of further ablation studies the issue that casts doubt on the novelty of our finding, or the fact that we work on improving EP? Or both?
> > >
> > > Further, we agree that it would be nice to better understand how exactly these tricks contribute to the "_trainability_" of these subnetworks. We believe that Fig 7 in our paper gives us some hints on the necessity of global pruning by comparing the layerwise sparsities of the subnetworks identified by different algorithms. For example, it seems plausible that the most effective way to distribute the "_sparsity budget_" is to spend it on the first and last layers, which is something that EP does not do since it enforces the same sparsity across all layers. We also find empirically that gradual pruning may not even be necessary if the regularization parameter is chosen extremely carefully. However, as we remark in Section 3, finding the appropriate regularization weight is quite challenging, especially to achieve a specific target sparsity. Thus, we find that gradual pruning is useful to ensure that the final subnetwork is of the target sparsity.
> > >
> > > Thank you once again for your time, and we appreciate the effort in engaging with us during this process.

---

### Official Review · Reviewer_BBr3 · 2022-07-15

**Rating:** 4
**Confidence:** 3
**Soundness:** 2 fair
**Presentation:** 3 good
**Contribution:** 2 fair

**Summary:**

This paper proposes an algorithm that finds sparse subnetworks at initialization, trainable to accuracy comparable or better than IMP with warm-up.

**Questions:**

If I understand the algorithm correctly, about half of the weights would be masked at initialization as score vector p is initialized randomly, which seems a little bit weird to me. According to Algorithm 1, only effective weights receive gradients, so if a weight is masked, it will be masked forever. So, could the authors explain the design behind this or correct my misunderstanding?

**Limitations:**

The authors address the limitations adequately.

**Strengths And Weaknesses:**

Strength:
This paper proposes an algorithm that finds winning tickets at initialization and validates the algorithm on several tasks.
The experiments are solid.
Weakness:
The novelty is limited.
The proposed method is a combination of different techniques.

---

> ### Author Response · Authors · 2022-08-02
> **Response to Reviewer BBr3**
>
> We thank the reviewer for acknowledging that we validate our algorithm on several tasks using thorough experiments.
>
> **`Limited novelty`**
> > Weakness: The novelty is limited. The proposed method is a combination of different techniques.
>
> **Our claim to novelty is not the algorithm (Gem-Miner), but the discovery of Rare Gems (per our title and abstract).** We establish the existence of lottery tickets in popular settings (CIFAR10/tinyImageNet, and others); these are setting where other algorithms fail sanity checks. In precisely this sense, we offer a novel finding; one that was sought after by several papers in the recent literature. Given modest familiarity with the _learning by pruning_ literature and the EP algorithm, Gem-Miner is indeed a relatively obvious modification (train scores bounded in $[0,1]$, quantize on forward, regularize and apply iterative freezing). This should not cast any negative light on what Gem-Miner discovers: trainable subnetworks at initialization, in popular settings, where other algorithms failed.
>
> **`Are weights masked forever?`**
> > If I understand the algorithm correctly, about half of the weights would be masked at initialization as score vector p is initialized randomly, which seems a little bit weird to me. According to Algorithm 1, only effective weights receive gradients, so if a weight is masked, it will be masked forever. So, could the authors explain the design behind this or correct my misunderstanding?
>
> As we discuss in Section 3, we compute gradients through the round function using the Straight-through estimator. This allows for non-zero gradients to be computed even when the weights are "pruned" and hence these scores continue to be updated. In fact, we observe empirically that some of these weights do come back over the course of training. We have added an additional clarifying statement regarding this in the revision.

---

### Author Response · Authors · 2022-08-02
**General comments to AC and All Reviewers**

We thank the reviewers for their constructive comments and excellent questions. We appreciate that all the reviewers acknowledge that we have "_extensive and thorough experimental results_" (Reviewer BBr3, Reviewer nnnE, Reviewer Loxt, Reviewer YCDn).

We are glad to hear that "_The results are strong and exciting, and also provide insightful analysis._" (Reviewer Loxt) and that "_The main reasons (i.e., global pruning and gradual pruning) for the effectiveness of the proposed method are well explained in the ablation study (Tab.3), which can be used as default setups in follow-up works._" (Reviewer YCDn).

We are heartened to see that "_Searching for subnetworks at initialization with better after-training performance is an important problem in the lottery ticket hypothesis (LTH). This paper proposes a new method to tackle this problem and achieves promising results._" (Reviewer YCDn).

As for the concerns/questions raised, we believe that we successfully addressed all of them sufficiently and reply in line to each review.

---

### Meta-Review · Area_Chair_jQW4 · 2022-08-20

**Recommendation:** Accept
**Confidence:** Certain

**Metareview:**

This paper proposes GEM-MINER to find lottery tickets at initialization. It tries to maximize the accuracy of subnetworks before weight training, and can discover subnetworks comparable to IMP with warmup while being much faster. The approach can pass the sanity checks proposed by prior work.

It received scores of 4568. All the reviewers agree that the problem studied in this paper is important for lottery ticket hypothesis (LTH). The reviewers appreciate the authors' extensive experiments and thorough analysis. Most of the concerns have been well addressed, though Reviewer BBr3 still showed concerns on why 'Rare Gem' can solve the problems others could not solve, and commented that more insightful discussions behind the finding are needed.

Overall, the AC thinks that this paper presents valuable findings, and the responses to reviewers' comments are adequate, therefore, the AC would like to recommend acceptance of the paper.

**Award:**

No

---

### Decision · Program_Chairs · 2022-09-14

Accept